# Promiscuous structural cross-compatibilities between major shell components of *Klebsiella pneumoniae* bacterial microcompartments

Lucie Barthe[1], Damien Balestrino[2], Bessam Azizi[1,3], Delphine Dessaux[1], Vanessa Soldan[4], Jeremy Esque[1], Thomas Schiex[3], Sophie Barbe[1], Luis F. Garcia-Alles[1]*

**1** TBI, Université de Toulouse, CNRS, INRAE, INSA, Toulouse, France, **2** Université Clermont Auvergne, LMGE, CNRS, Clermont-Ferrand, France, **3** MIAT, Université Fédérale de Toulouse, INRAE, ANITI, Toulouse, France, **4** Plateforme de microscopie électronique intégrative METi, Centre de Biologie Intégrative, CNRS, Toulouse, France

\* lgarciaa@insa-toulouse.fr

## Abstract

Bacterial microcompartments (BMC) are submicrometric reactors that encapsulate dedicated metabolic activities. BMC-H hexamers, the most abundant components of BMC shells, play major roles for shell plasticity and permeability. In part, chemical exchanges between the BMC lumen and the cellular cytosol will be defined by the disposition of amino acids lining the central BMC-H pores. Current models attribute to BMC-H a homo-oligomeric nature. The hexagonal symmetry of corresponding pores, however, would break down if hetero-hexamers formed, a possibility suggested by the frequent presence of multiple paralogs within BMC operons. Here, we gauged the degree of structural promiscuity between the 11 BMC-H paralogs from *Klebsiella pneumoniae*, a potential human pathogen endowed with the capacity to express three different BMC types. Concomitant activation of transcription of several BMC operons was first shown to be possible. By leveraging an adapted tripartite GFP technology, all possible BMC-H pair combinations were screened in *E. coli*. Multiple structural cross-compatibilities were pinpointed between homologs arising not only from the same BMC operon, but also from different BMC types, results supported by Alphafold and ESMFold predictions. The structural stability and assembly propensity of selected hetero-associations was established by biochemical means. In light of these results, we reinterpreted published lysine cross-linking mass spectrometry data to demonstrate that one of these hetero-hexamers, involving PduA and PduJ, was already detected to form in the shell of a recombinantly-expressed 1,2-propanediol utilization compartment from *Salmonella enterica*. Altogether, this study points to the need to embrace an augmented structural complexity in BMC shells.

**Data availability statement:** All relevant data are within the manuscript and its Supporting Information files.

**Funding:** The French National Research Agency (ANR) financially supported this work: ANR-19-CE09-0032-01. This work also benefited from a grant managed by the same agency, under the "Investissements d'Avenir" programme: ANR-18-EURE-0021. This work was granted access to the HPC resources of CALMIP supercomputing center. The funders had no role in study design, data collection and analysis, decision to publish, or preparation of the manuscript.

**Competing interests:** The authors have declared that no competing interests exist.

## Introduction

Bacterial microcompartments (BMC) are complex macrostructures composed of a protein shell encompassing an enzymatic set that defines its precise function. Polyhedral shapes have been inferred for several BMC types using techniques like electron cryotomography or transmission electron microscopy (TEM) [1–4]. BMC sizes spread from about 40 nm up till 600 nm [5,6]. Best characterized cases are the α- and β-carboxysomes, which contribute to $CO_2$-fixation by cyanobacteria and some autotrophs, or the 1,2-propanediol (PD) utilization (PDU) and ethanolamine (EA) utilization (EUT) compartments [7,8]. However, the catalog of BMC-mediated activities/functions is steadily expanding [9–11]. Indeed, bioinformatic surveys of available genomes uncovered BMC *loci* in about 20% of all sequenced bacterial genomes [11–13].

Compelling evidence supports that BMC confer competitive advantages to some bacteria under physiological contexts [14]. For instance, by processing EA or PD released in the gastro-intestinal tract, pathogenic microorganisms as well as some pro-pathogenic commensals, might out-compete other strains [15–17]. BMC would therefore contribute to boost the virulence of some pathogens [14]. Not surprisingly, non-pathogenic commensal strains have also developed BMC-related counteracting strategies [18].

Remarkably, about 22% of BMC-endowed organisms carry genetic information coding for two or more BMC types, with most extreme cases reaching up to 6 BMC loci [11]. A relatively well-characterized case is *Salmonella enterica LT2* (*Sal*), which contains both EUT and PDU compartments. Most often, the set of genes coding for shell components, auxiliary proteins and cargo enzymes are gathered in a single *locus* or operon. In some cases, the genetic information is split in several *loci*. This happens, for instance, in β-cyanobacteria, which distribute carboxysome genes in a main operon and satellite *loci* [8,12]. Shell proteins are often classified in three major classes: i) the most abundant subunits, stoichiometrically, adopt a PF00936 structural fold and associate as hexamers (BMC-H) [19]; ii) in the second group, each subunit is a fusion of two PF00936 domains. Consequently, the final pseudo-hexamer is in fact a trimer (BMC-T) [20,21]. BMC-H and BMC-T can be further subclassified depending on other criteria, such as the presence of permuted secondary structural elements, the existence of N-terminal or C-terminal extensions or the capacity to form hexagonal stacks [22,23]; iii) the components of the last group oligomerize as pentamers (BMC-P), each monomer obeying a PF003319 fold. Several lines of evidence [23–25], including high resolution structures of synthetic [26,27] and natural compartments [28] support a shell model with polyhedral facets and edges composed of BMC-H and, when present, BMC-T, while vertices are capped by BMC-P.

About 2.6 BMC-H, 0.9 BMC-T and 1.7 BMC-P gene paralogs are found on average per BMC *locus* [11]. These homologs generally share sufficient sequence identity as to envision that hetero-oligomers combining several homologs could form. Indeed, structural compatibilities were proved within a subgroup of BMC-H. CcmK3 and CcmK4 homologs from *Halothece sp.* PCC 7418, *Synechococcus elongatus*

PCC 7942 or *Synechocystis. sp.* PCC 6803 (*Syn6803*), as well as CcmK1 and CcmK2 from the last species, formed hetero-hexamers after recombinant co-expression in *E. coli* [29,30]. Exploring whether this is a widespread phenomenon is important, since heteromerization is anticipated to impact features like shell permeability [31–33], the capacity to recruit other BMC components [34,35], or the kinetics of compartment biosynthesis, degradation and repartition during cell division [36–38].

Herein, we selected for our investigation *Klebsiella pneumoniae* (strain 342, hereafter *Kpe*), an important human pathogen that harbors three different BMC operons (*eut1*, *pdu1a* and *grm2*), and demonstrated that hetero-hexamerization also concerns catabolic compartments. Transcriptomic data first supported that expression of several BMC might happen simultaneously. Subsequently, and mostly leveraging an adapted protocol to monitor protein-protein interactions (PPI) between BMC-H [39], based on the tripartite-GFP technology (tGFP) [40], we could pinpoint abundant structural compatibilities between BMC-H. A majority of combinations between the 11 *Kpe* BMC-H monomers indicated positive PPI, both when considering components from the same or from different BMC types. A systematic study of the different combinations with artificial intelligence (AI)-based multimeric structure prediction methods corroborated structural compatibilities. Finally, we reinterpreted reported cross-linking data [41], demonstrating the occurrence of hetero-hexamers in BMC shells. Our study thus extends previous conclusions established from studies of carboxysome components to other BMC types, arguing for the strong necessity to characterize hetero-oligomerization extent in natural hosts and to evaluate the consequences for BMC function and eventually for the organism pathogenicity/virulence.

## Results

### Genetic organization and regulation of Klebsiella pneumoniae BMC

Together with *K. quasipneumoniae*, and *K. variicola, K. pneumoniae* (*Kpe*) is responsible for many nosocomial and community-acquired pneumonia, urinary tract, and bloodstream infections in healthcare settings. *Kpe* harbors three different BMC *loci* (Fig 1). The first one closely resembles the *pdu1a* operon from *S. enterica* [42]. The signature pathway enzyme is the diol dehydratase (DDH) that converts PD to propionaldehyde and is composed of three subunits (PduC/PduD/PduE). The operon codes for 4 BMC-H (PduA/PduJ/PduK/PduU), 2 BMC-T (PduB/PduT) and 1 BMC-P (PduN). It

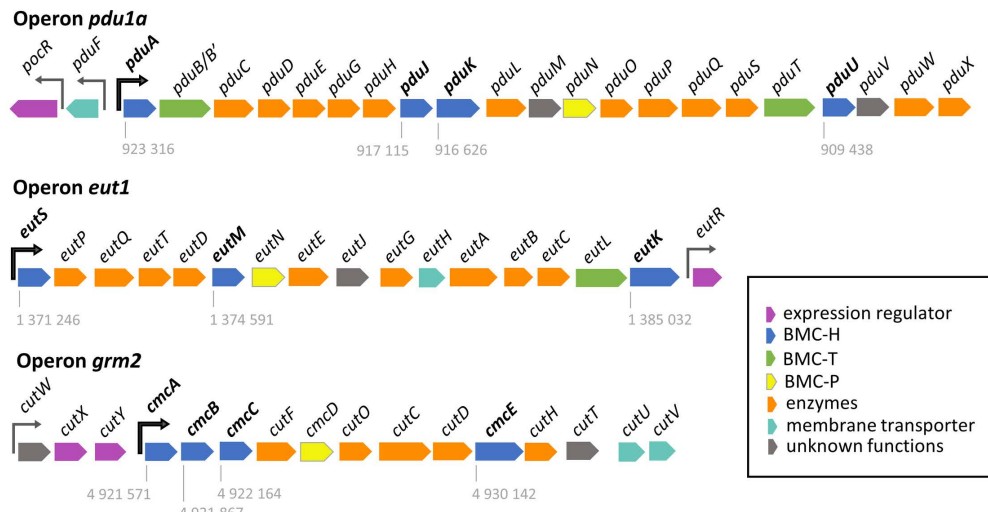

**Fig 1. Genetic organization of *Kpe* BMC operons.** The three BMC-coding operons of *Klebsiella pneumoniae* (strain 342) *are* shown, together with flanking regulatory elements. Promoters are represented by arrows, in black for those which directly control BMC transcription. Numbers beneath BMC-H coding sequences indicate the chromosomal location of the first nucleotide of the translation start codon.

is preceded by two adjacent and divergently transcribed genes, *pocR* and *pduF*, which code for a positive transcription factor directly acting on the *pdu1a* promoter and a PD transporter, respectively. The last two genes are followed by the *cob* operon that participates to the *de novo* synthesis of adenosyl-cobalamin (Ado-B$_{12}$), a cofactor required for PD and EA utilization and implicated in transcriptional regulation mechanisms [7,43,44].

The *eut1* operon also globally recapitulates the organization found in *S. enterica*. The key enzyme is the EA ammonia lyase, composed of EutB and EutC protein whose activity also depends on Ado-B$_{12}$. Among the 16 CDS that constitute the *Kpe eut1* operon, 3 of them correspond to BMC-H proteins (EutS/EutM/EutK), 1 to BMC-T (EutL) and 1 to BMC-P (EutN). As in *S. enterica*, the transcriptional regulator (EutR) is coded at the operon downstream end. *S. enterica* EutR was shown to activate the *eut* operon when both EA and Ado-B$_{12}$ were present [43]. Importantly, however, such activation was repressed by PD [42], something required to prevent the detrimental mixing of components from the two compartments. Accordingly, PD was preferentially consumed over EA as carbon source during growth.

*Kpe* is also endowed with a group-2 glycyl-radical enzyme-associated microcompartment (GRM2) that participates in choline (CL) metabolism. Its signature enzyme is the CL-trimethylamine (TMA) lyase, CutC, which catalyzes the decomposition of CL into acetaldehyde and TMA [10,45]. The corresponding *grm2 locus* of *Kpe* codes for 4 BMC-H (CmcA/CmcB/CmcC/CmcE) and a single BMC-P (CmcD). GRM2 are deprived of BMC-T[11,46]. Upstream the *grm2* promoter is found a small operon that codes for regulatory CutW-CutX-CutY. By similarity to what happens in uropathogenic *E. coli 536* strain [47], CutX and CutY are expected to activate *grm2* transcription in the presence of CL under anaerobic conditions.

## BMC transcription

The comparable organization of *eut1* and *pdu1a* operons in *S. enterica* and *Kpe*, with flanking *eutR* and *pocR* regulatory genes, suggested that similar transcriptional regulation schemes might apply. Thus, we investigated whether similar repression schemes operate in *Kpe*. Preliminarily, we sought for conditions in which this organism was able to grow on PD, EA or CL. Under aerobic conditions, EA could be utilized as carbon and nitrogen sources in minimal medium, provided that vitamin B12 was added, in agreement with the literature [15,43]. CL stimulated *Kpe* growth on minimal medium that also contained small amounts of yeast extract (YE, 0.1–0.2%). Something similar occurred under anaerobic conditions with EA or CL. However, no growth benefit could be attributed to the presence of PD, both under aerobic or anaerobic conditions and regardless of supplementation with diverse components: presence or absence of cyanocobalamin (vit-B$_{12}$), YE (0.1 to 0.5%), fumarate, amino acids, metal salts (including ferric citrate), or bile salts [16]. It is important to mention that metabolite utilization is not necessarily associated to BMC. This is illustrated by a recent report, which attributed EA utilization as carbon source under aerobic conditions to the main *Kpe eut* operon, while nitrogen utilization was basically dependent on a second short *locus* that includes EutB/EutC ammonia-lyase subunits but lacks genes coding for other EUT components, including those necessary for shell assembly [15]. Short locus enzyme sequences are 51–33% identical to those from the *eut1* operon. Similarly, PduC/PduD/PduE homologs are encoded in a small *locus* and exhibit even stronger resemblance to the PDU counterparts (75%/63%/61% identities). Two *cutC/cutD*-containing stretches are also present in *Kpe*, though more distant from GRM2 sequences (32–34% identities).

Taking into account such complexifying possibilities, we focused our investigation on the measurement of transcription shifts induced by PD, EA and/or CL. This was ascertained using reverse-transcription quantitative PCR (RT-qPCR) on total RNA extracted from cells growing under anaerobic conditions in NCE minimal medium supplemented or not with each individual metabolite (PD, EA or CL), or with combinations of two of them. We monitored upstream, middle and downstream regions from each operon (see M&M section). In that manner, a strong upregulation of the three *grm2* sections was induced by CL. Almost 500- and 100-fold higher transcript levels were measured for *cmcA* or *cutC*, respectively (Table 1, see also S1-S2 Tables). GRM2-induction by CL was repressed by EA, although not completely. PD addition was without effect. To our surprise, none of the screened *eut* or *pdu* transcript levels was significantly modified by the

**Table 1. Induction of *Kpe* BMC transcription in response to metabolite presence.**

| Subs. | *cmcA* | *cutC* | *cmcE* | *eutS* | *eutM* | *eutK* | *pduA* | *pduJ* | *pduU* |
|---|---|---|---|---|---|---|---|---|---|
| **EA** | 1.0 | 0.4 | 0.2 | 1.1 | 3.8 | 0.4 | 0.1 | 0.3 | 0.9 |
| **PD** | 0.4 | 2.4 | 0.6 | 0.7 | 2.2 | 1.0 | 0.8 | 3.8 | 5.7 |
| **CL** | 477 | 76 | 23 | 0.7 | 17 | 1.3 | 5.1 | 36 | 27 |
| **EA+PD** | 0.8 | 0.3 | 0.2 | 0.6 | 9.2 | 0.3 | 1.5 | 0.4 | 0.8 |
| **EA+CLª** | 18 | 2.8 | 1.0 | 13 | 478 | 5.7 | 0.6 | 2.7 | 1.7 |
| **PD+CL** | 603 | 67 | 17 | 0.7 | 23 | 1.0 | 19 | 79 | 47 |

Transcription levels are calculated from three independent replicates, and are given as fold changes relative to the condition without substrate. ª These series of experiments manifested a strong variability among replicates. Statistical information and measurements on individual replicates and house-keeping genes are given in S1 and S2 Tables, respectively. See also Materials and Methods for further information.

corresponding substrate, the highest upregulation being a highly dispersed 6-fold increase measured for *pduU*. Only CL continued to promote higher transcript levels, as estimated for *eutM* (17-fold), *pduJ* (36-fold) and *pduU* (27-fold). Intriguingly, EA or PD promoted a clear augmentation of transcript levels of their cognate BMC genes when combined to CL (a *ca*. 500-fold strongly dispersed average for *eutM* or a 80- and 50-fold for *pduJ* or *pduU*, respectively). Noticeably, when CL and EA were supplemented together, values suffered from a considerably high dispersion, something particularly evident for *eut* genes. Experimental manipulation errors were ruled out by correct readings on *proC* and *rpoD* housekeeping gene transcripts. Variability deriving from inadequate primer selection was also excluded as amplification reactions of the same stretches were reproducible for cultures grown in the presence of other substrates.

Globally, these data supported the possibility that components from different BMC can be expressed simultaneously in *Kpe* cells.

## Kpe BMC-H sequence and structural predictions

Sequence considerations permitted to classify the 11 *Kpe* BMC-H into three groups (S1A Fig). The first one includes all canonical BMC-H: CmcA, CmcB, CmcC, EutM, PduA and PduJ. Identities range within this group between 56% (CmcA *vs* EutM) and 95% (CmcA *vs* CmcB), and are even higher (60–98%) when considering only residues at the interface between monomers (S1B Fig). The second group comprises BMC-H with long C-terminal extensions (40–70 residues) compared to canonical BMC-H. This group includes CmcE, EutK and PduK. Their sequences show greater divergence from each other than from the canonical BMC-Hs within their respective BMC. Finally, EutS and PduU, which belong to the third BMC-H group, carry circular permutations of secondary structure elements, compared to canonical proteins. They also possess N-terminal extensions that associate in the context of a hexamer to build a narrow β-barrel above the central pore, as demonstrated in crystal structures for other homologs [48–50]. As already known [11], these proteins have more in common with each other (56% identity) than with homologs of their respective BMC (19–25%). The difference becomes more pronounced when only interfacial residues are considered (73% for EutS-PduU, as compared to 15–27%, S1B Fig), which suggests that different evolutionary constraints might apply on this BMC-H group.

The potential formation of *Kpe* BMC-H homo-hexamers was explored with two AI-based multimeric structure prediction methods: AlphaFold2-Multimer (AF2) [51], which primarily infers residue-residue relationships through multi-sequence alignments (MSA), and ESMFold [52], which predicts structures directly from amino acid sequences using a transformer-based language model, ESM-2. The two algorithms proposed homo-hexamers for canonical BMC-H, each monomer adopting the characteristic PF00936 fold (S3 Table, S2A-B Fig). Prediction quality scores pointed to high-confidence associations [global and interface predicted local difference distance test (pLDDT) > 75, interchain and interface predicted aligned error (PAE) < 3.5 and AF2-multimer global or interface predicted template modelling

scores (pTM and ipTM) above 0.7, see M&M for the detailed description of prediction quality metrics]. These values were similar to those obtained in predictions for the positive control based on the single BMC-H from *M. smegmatis* (RMM), and differed from data for negative controls based on the monomeric buckwheat trypsin inhibitor (BWI) or the insoluble *Syn6803* CcmK3. In fact, AF2 metrics suggested homo-hexamer formation for the latter, pointing to a possible bias caused by the presence of similar sequences that would impact the MSA and eventually the confidence scores and the comparison between models of distinct proteins. Interaction energies (ΔE) estimated for relaxed structures were also in the range of those calculated for RMM and available crystallographic structures of BMC-H homo-hexamers (S4 Table).

Models predicted by AF2 for BMC-H from the second group, CmcE, EutK and PduK, were associated with relatively low-confidence quality scores, though ΔE seemed correct. Since low quality metrics for CmcE, EutK and PduK might be caused by the long C-terminal flexible extensions, which were predicted with large unstructured parts by both methods (S2 Fig), we also evaluated PAE values for only residues from the PF00936 core (c_PAE), which confirmed high confidence models for CmcE and EutK. Conversely, ESMFold directly rejected the oligomerization of PduK, and pointed to unlikely CmcE and EutK hexamer occurrence, even when considering only core residues. pLDDT and PAE indicators were in the range of those for negative controls. Remarkably, the two methods predicted the presence in C-terminal extensions of a small structured Cys-rich motif in PduK, which could harbor a [Fe-S] cluster domain (S2E-F Fig), something partly demonstrated for the *S. enterica* homolog [53], and a fold in EutK typical of nucleic acid-binding proteins (PF16365) (S2I-J Fig), as proposed for the monomeric *E. coli* homolog too [48]. Indeed, high confidence AF2 scores (pLDDT between 70 and 90) are attributed to these portions in models deposited at the EMBL-EBI site (EutK entry B5XVQ9; PduK entry B5XUS7).

Predictions for EutS and PduU diverged considerably depending on the AI-algorithm. ESMFold indicated an absence of oligomerization for EutS or a hexamer model with low-confidence metrics for PduU. The disposition of the first β-strand of the EutS monomer (corresponding to the canonical β4, S1A Fig) deviated in fact from the PF00936 fold (S2D Fig), adopting a conformation that would collapse with a neighbor monomer if a hexamer formed. ESMFold also attributed helical structures to a portion of the EutS and PduU N-terminal extensions, in disagreement with expectations based on the known structures of PduU and EutS homologs [48–50]. Conversely, AF2 predicted for both proteins high-confidence homo-hexamers displaying favorable binding energies (S2C Fig) and N-terminal extensions reproducing faithfully the short β-barrels modelled in crystal structures.

## Experimental structural viability and oligomeric state of recombinant Kpe BMC-H

The experimental behavior, oligomerization state and assembly properties of individual *Kpe* BMC-H was next inspected. SDS-PAGE analysis confirmed that all 11 His$_6$-tagged proteins could be over-expressed in BL21(DE3) cells (Fig 2). Only EutM was produced in moderate yields. After cellular lysis and centrifugation, most of them were retained in pellets, pointing to protein aggregation or assembly into supramolecular structures. As exceptions, EutM and CmcB remained soluble and could be purified directly. All other BMC-H could be recovered from pellets after protein disassembly with 1 M urea at 4 °C (which does not permit resolubilization of aggregated proteins). Corresponding purified fractions, thus in the absence of urea, remained soluble.

Urea purified His$_6$-tagged BMC-H were analyzed by size-exclusion chromatography (SEC-HPLC). EutM and CmcB samples purified from first soluble fractions, or the two purified EutK fractions were also chromatographed. Species eluting at volumes expected for hexamers were revealed for a majority of cases (S3A Fig and S5 Table). Exceptions were PduK and EutK, both behaving as monomers, as predicted by ESMFold (AF2 also failed for EutK). Indeed, EutK eluted at a time expected for a protein approximately half its size (9 kDa) when the first purified fraction was injected, while estimations were closer to a monomer size (17 kDa) for the urea-recovered sample, something unexpected considering their similar SDS-PAGE migration (Fig 2). Highest intensity peaks with CmcE (both N- or C-ter His$_6$-tagged) and PduU occurred at

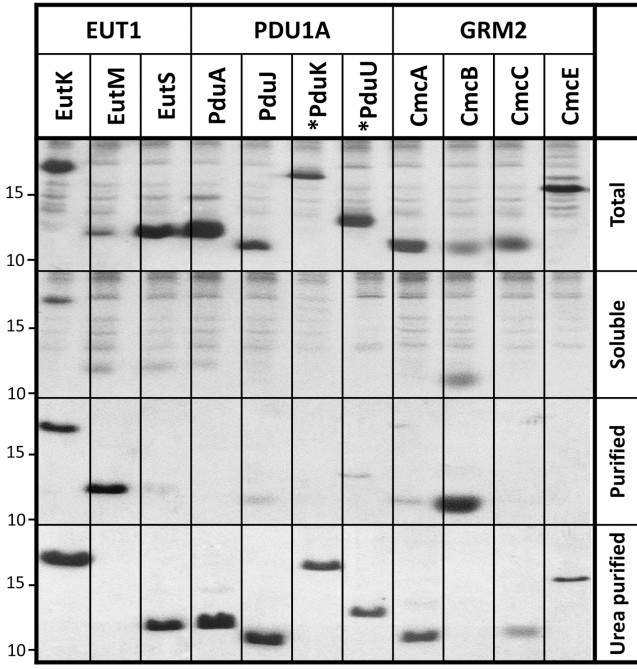

**Fig 2. Expression of *Kpe* BMC-H in *E. coli*.** SDS-PAGE characterization of His₆-tagged *Kpe* BMC-H over-expressed in BL21(DE3). From top to bottom are presented total cellular fractions, soluble fractions retained in supernatants after lysis and centrifugation, and material purified by cobalt-based affinity chromatography. The bottom panel corresponds to fractions that were purified similarly after load into the same affinity resins of material resolubilized from pellets treated with 1 M urea. Only the portion of the gel showing the proteins of interest is shown. An asterisk preceding the protein name is to denote tag attachment to the protein N-terminus; C-terminal tags are present for all other cases. The total and soluble fractions, likewise purified and urea-purified fractions, allow a direct comparative view of protein levels since gel loads were identical while fraction volumes remained constant throughout the experiment. Note however that Coomassie staining/destaining might slightly vary from gel to gel. The approximate migration of protein ladder species of 10 and 15 kDa sizes is indicated on the left side. See M&M for further details.

volumes expected for aggregated/assembled species (> 500 kDa), hexamer-compatible peaks being faint. High molecular-weight species were also abundant in EutM, EutS, PduA, and PduK samples, which suggested that urea treatments did not fully dismantle the pelleted macro-assemblies.

BMC-H assembly propensity was inspected by TEM after over-expression in *E. coli* (Fig 3). Only the three BMC-H with long C-ter tails failed to reveal structured organizations. Four cases, EutM, PduA, PduJ and CmcC, resulted in mixtures of varied motifs, some compatible with piled sheets (stripped motifs with variable separations in the 3–8 nm range), others with nanotube bundles (16–24 nm interspacing) that resembled motifs reported for PduA from *S. enterica* or *Cit. freundii* [54,55]. Nanotubes were wider (20–43 nm) for PduJ, which also displayed piled and rolled sheets (8–12 nm spacing) and potential hollow spheroids (60–80 nm). Possible consequence of its low expression level, EutM nanotubes were less evident than for the other 3 BMC-H, appearing in the center of cells as short rods (Fig 3B). Finally, EutS, PduU and CmcB expressing cells displayed patches that looked like fingerprints: 4–6 nm periodicities for EutS and 5–6 nm for CmcB and PduU. CmcA structures were less clear, possibly combining different kinds of motifs.

Overall, these data indicated that *Kpe* BMC-H are correctly produced in *E. coli* and form oligomers that further assemble into higher order supramolecular organizations. Indirectly, the occurrence of organized assemblies supported oligomerization for otherwise doubtful cases like PduU. On the contrary, EutK monomeric behavior and failure to assemble seemed in conflict with a retention in cellular pellets. The apparent discrepancy might stem from interactions with other sedimented components, such as DNA.

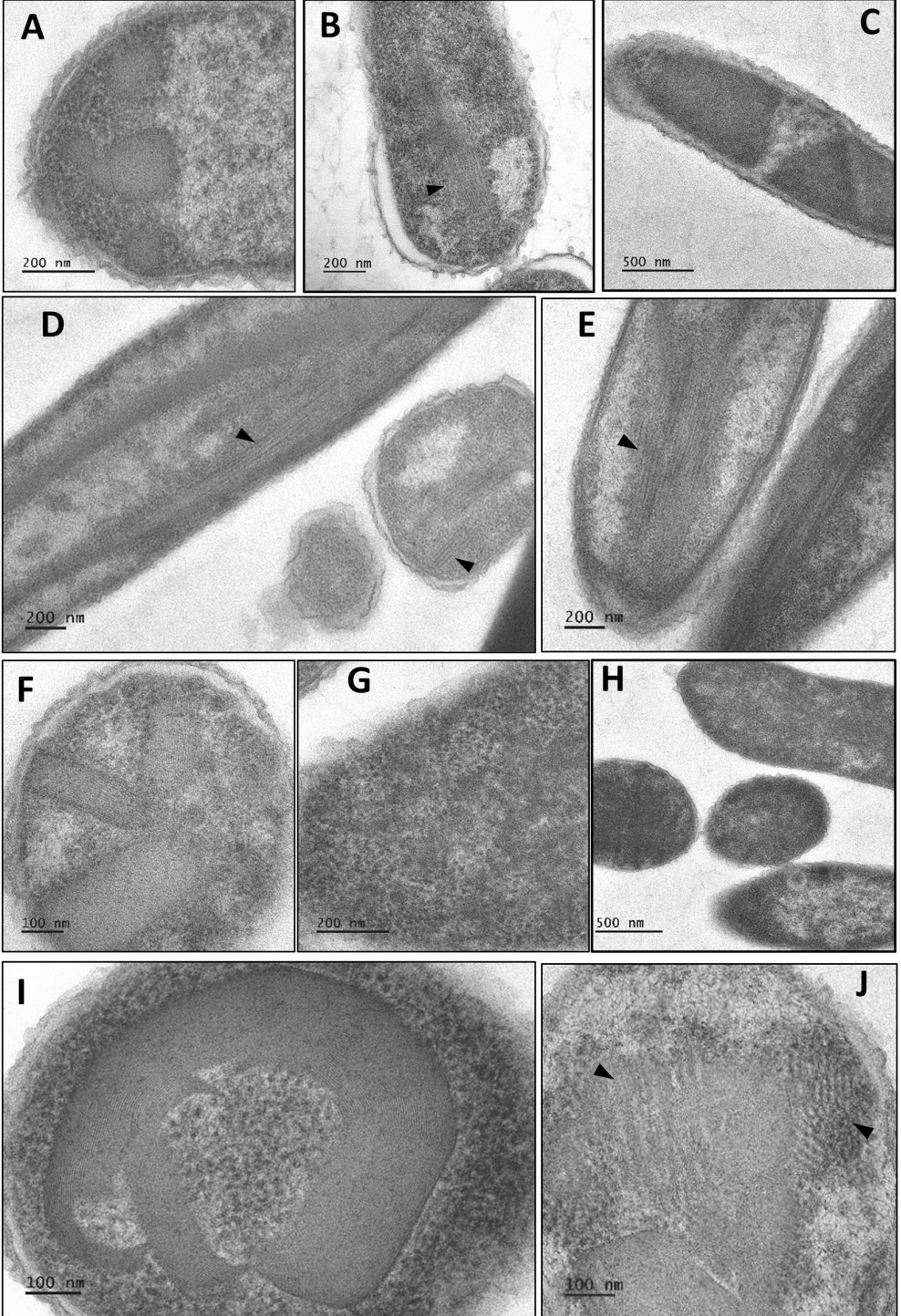

**Fig 3. Electron microscopy of *Kpe* BMC-H homomers.** *E. coli* BL21(DE3) cells over-expressing the following BMC-H were imaged by transmission electron microscopy: EutS **(A)**, EutM **(B)**, EutK **(C)**, PduA **(D)**, PduJ **(E)**, PduU **(F)**, CmcA **(G-H)**, CmcB **(I)** and CmcC **(J)**. All BMC-H carried C-terminal His$_6$ tags, except for PduU (N-terminal). The arrows point to motifs interpreted to be nanotubes.

## The tripartite GFP as tool to study BMC-H oligomerization

Our major purpose was to characterize potential interactions between *Kpe* BMC-H, as a means to anticipate the occurrence of non-canonical hetero-combinations in the natural host, and, by extension, in other microorganisms harboring multiple BMC-H paralogs. Our screen was based on the tGFP technology [40], since it is amenable for high-throughput screening of protein interactions and well-suited for the study of BMC shell proteins [39]. All possible paired combinations between the 11 *Kpe* BMC-H were covered. Briefly, necessary DNA fragments coding for a first BMC-H in fusion to the GFP10 β-strand and a second BMC-H connected to the GFP11 β-strand were mounted in a pET-based expression vector that also coded for the remaining part of GFP (GFP1–9 fragment) (S4A Fig). Subsequent GFP reconstitution and fluorescence signal boosting is expected if the two proteins associate (S4B Fig). Bearing in mind that tags often impact expression levels and could interfere with interactions, all the 8 possible tagging attribution/orientations were considered: both GFP10 and GFP11 in fusion to each protein C-terminus (C/C), N-terminus (N/N) or the two other combinations (C/N or N/C). This led to a total of 484 plasmid constructs. Forty-four were for combinations of GFP10 and GFP11-fusions of the same BMC-H, each one with the 4 different tag orientations (11 homo-pairs). The remaining 440 cases covered all 55 possible combinations of different BMC-H (hetero-pairs), each case comporting 8 combinations that included different tag orientations and exchange of the identity of the GFP10- and GFP11-carrying BMC-H. Maximal fluorescence readings ($F_{max}$) averaged over two independent experiments (comprising > 6 clones) are represented in Fig 4 (numerical values with standard deviations, as well as the estimated times of half-fluorescence are provided in supplementary S1 File).

The combinations of homo-pairs were first considered (diagonal of Fig 4), as fluorescence readings could be confronted with data presented above. To facilitate the visualization of effects of tag orientation, $F_{max}$ were also plotted (Fig 5A and S5 Fig). Globally speaking, the highest signals were detected with C/C orientations. Within this configuration, canonical BMC-H resulted in comparable (CmcC and EutM) or considerably higher fluorescence (CmcA, CmcB, PduA and PduJ) than the reference case, which combined two copies of RMM with GFP10 and GFP11 tags in C/C orientation. Similar values were also recorded for CmcE. On the contrary, PduU fluorescence was under the negative signal threshold established by two controls that corresponded to homo-pair combinations of either CcmK3 from *Syn6803* or BWI. $F_{max}$ values with EutK, EutS and PduK fell between those for RMM and negative controls. The introduction of N-tags progressively induced a decline of fluorescence with all BMC-H (S5 Fig), suggesting that either expression levels or the screened interaction might be impacted by the tag.

To better understand these data, the expression and solubility of individual BMC-H with GFP10 or GFP11 tags at N- or C-terminus were monitored (S6 Fig). Excluding CmcC, BMC-H band intensities in cellular and soluble fractions looked similar, indicating that GFP10/GFP11 constructs might be less prone to form macro-assemblies than $His_6$-tagged constructs. N-ter tagging frequently led to reduced expression levels, especially with the GFP11. In spite of such differences, weak $F_{max}$ readouts with EutK, EutS and PduU could not be justified by weak protein levels, instead supporting oligomerization failure. Of note, smears of lower sized bands were noticed for EutS, PduU, and PduK samples, suggesting proteolysis. With the latter, this might reveal auto-proteolytic damage caused by the predicted Fe-S cluster-harboring domain. On the opposite side, CmcE oligomerization was supported by the comparatively high $F_{max}$ values despite faint protein levels.

In summary, the tGFP screen correctly informed on homo-oligomer formation by CmcA, CmcB, CmcC, CmcE, EutM, PduA and PduJ, as well as on non-associating cases EutK and PduK, but failed to detect EutS and PduU oligomerization characterized here for the two $His_6$-tagged BMC-H, and reported for other homologs. The overall data also indicated that AF2 and ESMFold perform well as predictive tools. The quality metrics and interaction energies of proposed models were consistent with the majority of canonical BMC-H experimental behaviors (S3 Table). AF2, but not ESMFold, also agreed with observation for EutS and PduU. However, predictions of hexamer formation for BMC-H with long C-terminal extensions most often seemed to fail, both when using AF2 (EutK, PduK) or ESMFold (CmcE and EutK).

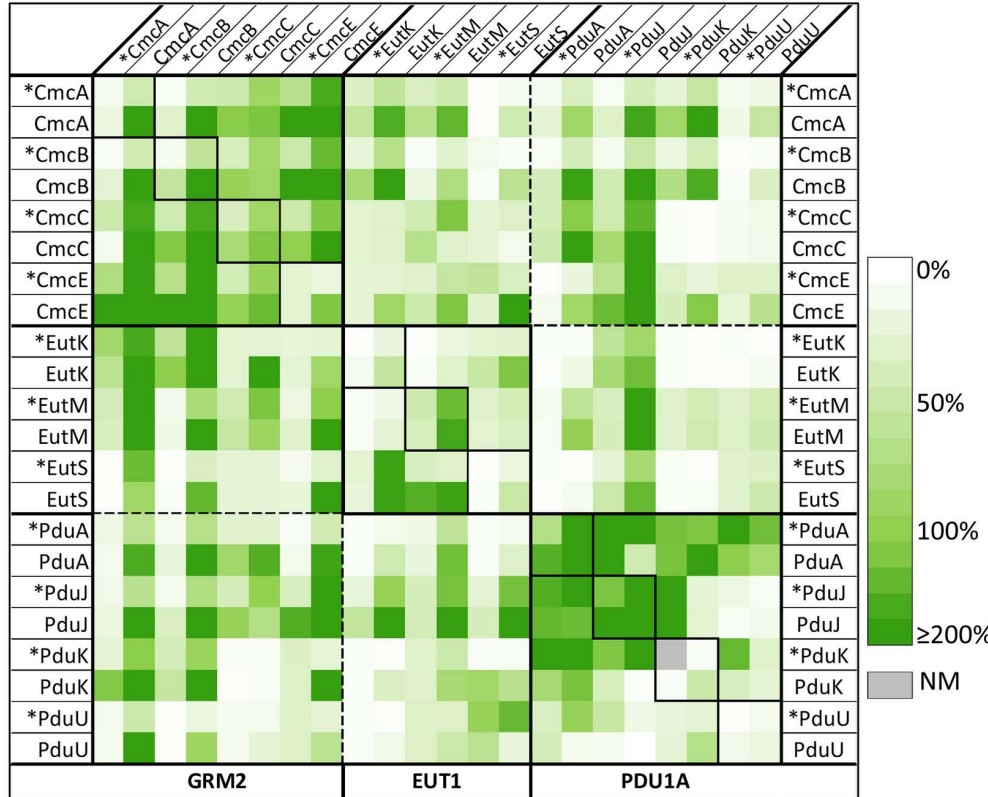

**Fig 4. Relative F<sub>max</sub> measured in tGFP assays for *Kpe* BMC-H pairs.** Screens consisted in a combined expression of a first BMC-H partner in fusion to the GFP10 (indicated horizontally at the top) and a second BMC-H tagged with the GFP11 (vertical labels), together with the GFP1-9 necessary for fluorescence reconstitution. Tags were either at the N- terminus of BMC-H (indicated by the asterisk), or at C-terminus (all others). $F_{max}$ values expressed as percentages of the RMM/RMM reference case (100%) are represented following the colour scale shown on the right. Each represented value corresponds to the average of at least 6 measurements, generated in 2 independent experiments. Standard deviations are provided in supplementary S1 File. NM: not measured, as the corresponding plasmid could not be prepared.

## Abundant structural compatibilities between BMC-H from the same BMC

Fluorescence readings for hetero-pairs that combined BMC-H coming from the same BMC pointed to abundant cross-interactions (Figs 4 and 5B). Structural promiscuity would be higher within the GRM2 and PDU shells than between EUT BMC-H, as can be visually inferred from data represented in Fig 4. EUT BMC-H combinations were also more sensitive to tag orientations. Some heteromeric associations could be expected on the basis of a high sequence resemblance. Our data supported that this happened for combinations of CmcA, CmcB and CmcC homologs, also between PduA and PduJ. Some other pinpointed potential interactions were more surprising. For instance, PduK (N-oriented) resulted in high $F_{max}$ values when combined to PduA or PduJ. In fact, according to tGFP data, PduA would be able to associate with all homologs from the PDU1A, in agreement with a proposed role as protein interaction hub [13]. Less clear, PduK/PduU led to positive signals in only one of the eight screened configurations. Concerning the GRM2 group, a noticeable result was the strong $F_{max}$ augmentation for CmcE combinations with CmcA or CmcB, when compared to the CmcE homopair, something that was reproduced in 6 of the 8 configurations of each hetero-pair. In fact, strong fluorescent signals occurred with all GRM2 BMC-H combinations, hinting at a remarkable promiscuity and raising the question of whether oligomers composed by 3 or 4 BMC-H homologs might occur in GRM2 shells. Also unanticipated, EutK/EutS fluorescence (C/C orientation) was considerably stronger than when screening the respective homo-pairs. Something similar

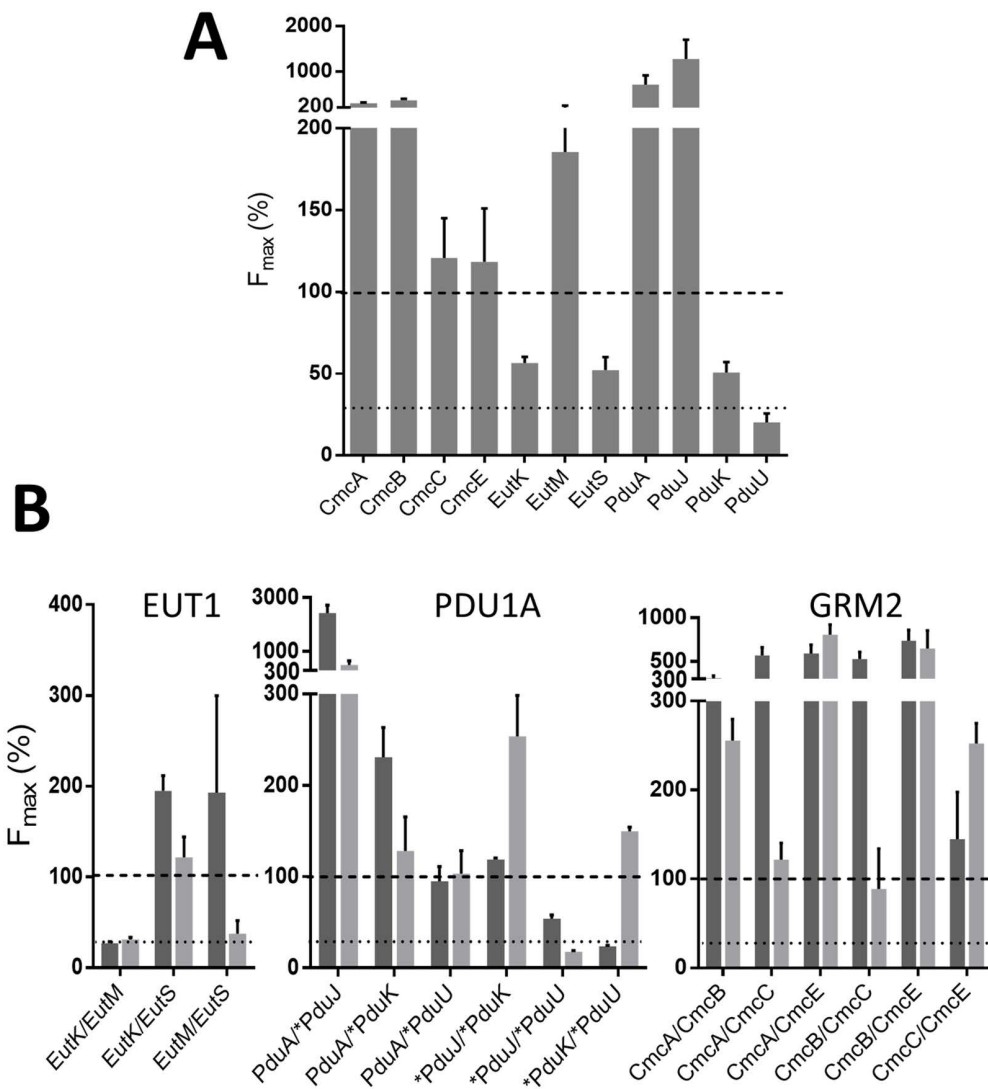

**Fig 5. Monitoring interactions between *Kpe* BMC-H couples. *A* –** tGFP reconstitution fluorescence signals were recorded for the indicated *Kpe* BMC-H, constructed as homo-pairs carrying GFP10 and GFP11 tags on the C-termini (C/C). Data recorded in similar experiments for other tag configurations are presented in S5 Fig. ***B* –** Protein-protein interactions between BMC-H from the same BMC. Studied hetero-pair combinations are indicated using a BMC-H1/BMC-H2 format below the X-axis. $F_{max}$ values plotted here are from screens of either BMC-H1-GFP10/BMC-H2-GFP11 (dark grey) or BMC-H2-GFP10/BMC-H1-GFP11 (light grey) combinations. For the two panels, maximal fluorescence signals ($F_{max}$) are expressed as a percentage of the value measured for the RMM homo-pair with C/C orientation (dashed line). The dotted line represents the highest threshold signal measured for negative-cases based on homo-pair combinations of either CcmK3 from *Syn6803* or of BWI. See Fig 4 for additional details, as data plotted in this figure correspond to a selection of those presented in that figure.

was obtained for EutS/EutM combinations, though the result was strongly impacted by the inversion of the BMC-H in fusion to GFP10 or GFP11.

Many of the hetero-associations pointed out by the tGFP data appeared to be supported by AF2 and ESMFold, which globally concurred to similar results (S6 Table). Thus, associations with alternating monomers (organization mode ABABAB, A and B representing the two monomer types) were invariantly recovered as top ranked models, with the exception of all combinations including EutS or PduU (excluding AF2-prediction for PduK/PduU), which resulted in models

where each type of monomer clustered together (AAABBB) and pTM and ipTM below acceptable thresholds, overall hinting at a lack of cross-interaction for these cases that is in partial disagreement with tGFP measurements. The two approaches validated a structural viability for pairs of canonical BMC-H (CmcA/CmcB, CmcA/CmcC, CmcB/CmcC, and PduA/PduJ), which resulted in quality metrics and binding energy scores (ΔE) similar to values obtained for a positive control based on the combination between CcmK1 and CcmK2 from *Syn6803* (S6 Table) or to estimations based on crystal structure models (S4 Table). ESMFold also supported interactions involving BMC-Hs from the first and second groups (CmcA/CmcE, CmcB/CmcE, CmcC/CmcE, EutK/EutM, PduA/PduK and PduJ/PduK), which, excluding EutK/EutM, caused high tGFP $F_{max}$ readings. Conversely, core_PAEs associated to similar ABABAB predictions with AF2 remained exceedingly high [please note that the quality metrics scores seem to differ depending on the method, as higher (better) pLDDT values in AF2 predictions were not often accompanied by lower (better) PAE values, when compared to ESMFold metrics, altogether arguing for a need to apply different thresholds depending on the algorithm]. Interestingly, the two algorithms suggested that formation of hetero-hexamers between CmcE and CmcA/B/C monomers might be reinforced by a disulfide bridge between the CmcE Cys80 and the Cys24 of the other homologs (S7 Fig).

## Co-purification experiments validate tGFP-unveiled cross-interactions

Like any other approach intended for mapping PPI interactions [56], the tGFP was presumed to rise both false-positive and false-negative readouts. In this section, we verified some selected combinations using a different biochemical approach, similar to the one applied in previous studies [29,30]. Briefly, the first BMC-H was connected to a FLAG peptide, while the potential partner was His$_6$-tagged. The FLAG construct should co-purify in association with the His$_6$-tagged protein in cobalt-based affinity resins provided that their interaction/association is strong enough as to endure necessary treatments and column washing steps. The following cases were selected among those revealing a potential positive interaction in the tGFP assay: EutK/EutS, EutM/EutS, PduA/PduJ, PduJ/PduK, PduU/PduK, CmcA/CmcC, CmcE/CmcA and CmcB/CmcE. Two presumed non-interacting cases, EutK/EutM and PduU/PduJ, were also included.

Good expression in *E. coli* and solubility of the two partners occurred for the majority of combinations (Fig 6A). The presence of two bands in purified fractions directly supported cross-associations between PduA/PduJ, PduJ/PduK, CmcE/CmcA and CmcB/CmcE, something confirmed by western blots (WB) using a FLAG-specific antibody (Fig 6B). WB analysis also permitted to confirm the CmcA/CmcC interaction. Less clear-cut was EutS/EutM, which bands could not be resolved and WB pointed to low level of co-purification of EutS-FLAG. Formation of EutK/EutS was not confirmed under this assay. Purification led to an enrichment of EutS-His$_6$, whereas EutK-FLAG bands were practically absent, contrasting with an inversed intensity ratio detected in the total cellular or soluble fractions. This also happened with EutK/EutM, which was confirmed not to interact, as indicated by the tGFP assay. PduU/PduK and PduU/PduJ combinations gave unclear results, observations being strongly influenced by the supplementation of β-mercaptoethanol (βME, 1 mM) prior to purification. Namely, the two partners were clearly visible in purified fractions in the absence of βME (S8 Fig), but were lost under reducing conditions (though still revealing surprisingly strong signals in WB, Fig 6B). Although unspecific binding of PduU driven by cysteine residues was suspected, we could not conclude on whether PduU/PduK or even PduU/PduJ interacted together or not.

Comparison of SEC-HPLC elution profiles with those observed for respective homo-pairs directly supported some cross-interactions (S3B Fig). Most remarkable, PduJ/PduK eluted with an apparent molecular weight (MW) of 110 kDa, clearly above species characterized for individual partners (S5 Table). Something similar occurred for CmcA/CmcE, CmcB/CmcE, CmcA/CmcC and PduA/PduJ. EutK/EutS and EutS/EutM resulted in HPLC profiles with minor peaks detected within the 5–500 kDa resolving range of the SEC-column. Purified samples involving combinations with PduU, and EutK/EutM, gave peaks exclusively at MW higher than 500 kDa.

A notable difference between data collected for homo- or heteromers concerned the distribution of BMC-H between soluble and assembled pools. With the exception of PduA/PduJ, the expression of hetero-pairs resulted in materials that

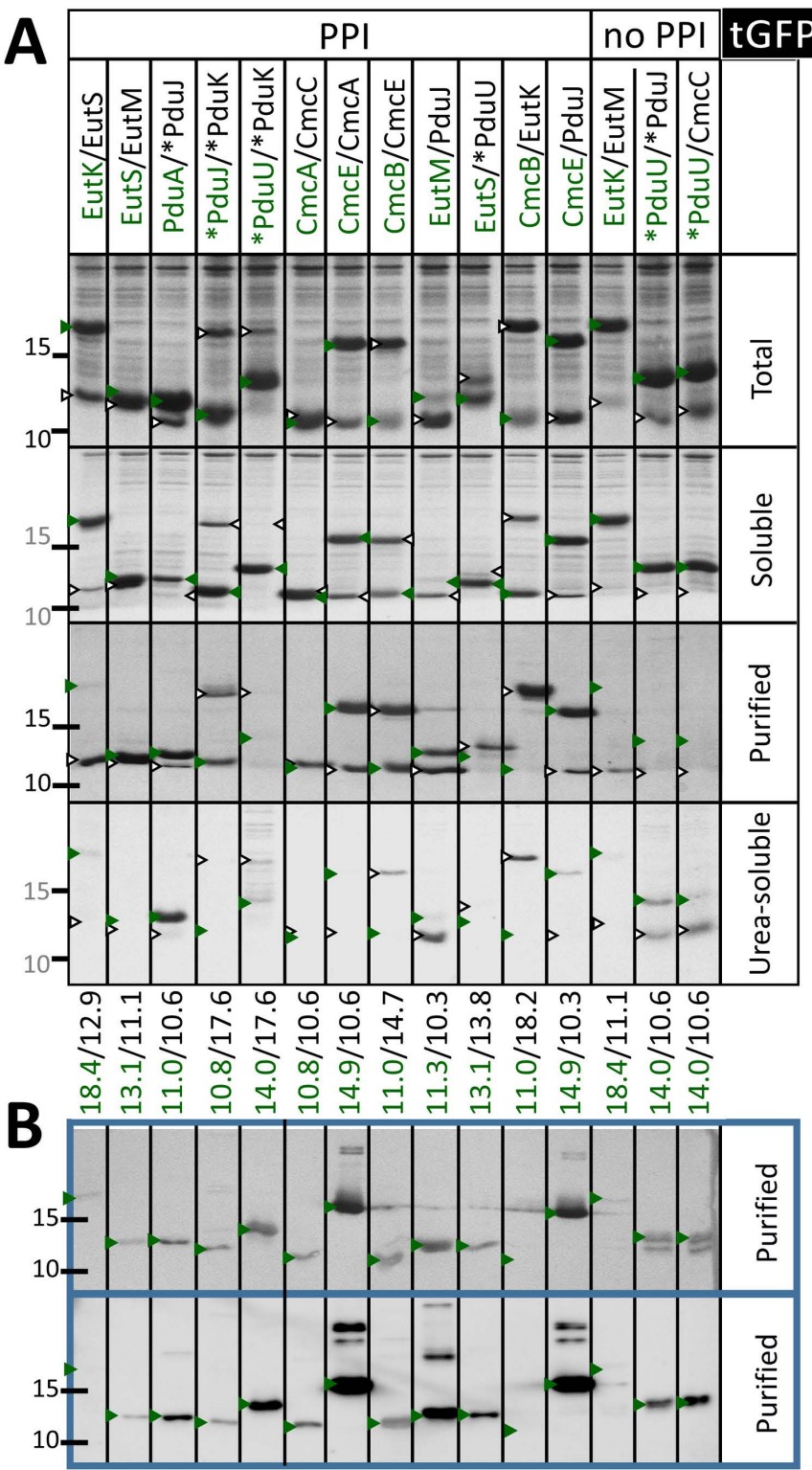

**Fig 6. Verification of *Kpe* BMC-H heteromerization. A.** Characterization by SDS-PAGE of fractions collected from *E. coli* cells over-expressing indicated *Kpe* BMC-H couples. The first component is FLAG-tagged (green label), second His₆-tagged (black). Soluble fractions were supplemented with 1 mM β-mercaptoethanol prior to loading into purification resins. Green and black arrows indicate the approximate migration position of each

partner, theoretical molecular weights being given at the bottom of the panel, following the same color code. Other details are as in Fig 2. **B.** Detection of co-purified FLAG-tagged species by western blot (WB). Two development techniques were applied: based on alkaline phosphatase reaction on (top) or HRP reaction on a chemiluminescent substrate (bottom), as explained in M&M.

remained basically soluble despite high expression levels, contrasting with data for homo-pairs, where a majority of cases was found abundantly retained in the pellet (Fig 6 *vs* Fig 2). PduA/PduJ assembly was confirmed by TEM (S9 Fig). Motifs recapitulated arrangements formed by individual proteins. Nanotube bundles displayed however sharper straight edges than those visualized for individual PduA or PduJ. As a counter-example, CmcA/CmcC over-expressing cells did not reveal organized structures (S9B Fig). The absence of sedimentation for most cases might be considered as symptomatic of the occurrence of cross-associations (or structural interferences), in view of strong assembly tendencies of homomers. It might also denote an impossibility of heteromers to build higher order structures.

### Structural compatibilities between BMC-H from different BMC types

Fluorescence measurements presented in Fig 4 also suggested that BMC-H from different compartments might cross-interact together. According to tGFP signal levels, 12 out of 16 possible hetero-pair combinations would be positive when combining BMC-H from the GRM2 and PDU1A groups, 10 over 12 for GRM2/EUT1 combinations and 5 over 12 for PDU1A/EUT1 (see also S10 Fig). Again, all cases mixing canonical BMC-H conducted to high $F_{max}$, once more confirming the high degree of structural promiscuity within this particular BMC-H cluster. A few combinations between canonical proteins and BMC-H with long C-terminal extensions were revealed highly fluorescent too (*e.g.,* CmcB/EutK, CmcA/PduK, CmcE/EutS or CmcE/PduJ). Co-purification experiments with FLAG/His$_6$ constructs clearly validated two of the three screened interactions within these BMC-H groups (EutM/PduJ and CmcE/PduJ, but not CmcB/EutK) (Fig 6). The latter also eluted in SEC-HPLC experiments as a single peak, attributed to EutK monomers (S5 Table). Worth-mentioning, EutS/PduU gave a comparable $F_{max}$ to the RMM reference, which contrasted with readings for EutS or PduU homo-pairs (Fig 5A). However, the interaction remained less certain in co-purification experiments. Despite the fact that the two components seemed correctly expressed and soluble, the EutS-FLAG and His$_6$-PduU partners eluted together only in the absence of βME (Fig 6 and S8 Fig). Nonetheless, FLAG peptides were detected with significant intensities in WB for the two experiments. Irrespective of βME presence, only high MW species were detected in SEC-HPLC, resembling PduU chromatograms (S5 Table).

AF2 and ESMFold predicted ABABAB hetero-hexamers for all 11 combinations between canonical BMC-H, in good agreement with empirical observations. ABABAB models were also generated for 15 remaining possible combinations between members of the two first BMC-H groups (S7A-S7B Tables), rising a notable discordance for weakly fluorescent combinations EutK/PduA, EutK/PduK, EutM/PduK, and CmcC/PduK (S10 Fig), though EutK/PduK pTM and ipTM scores were below confidence threshold. Quality scores accompanying some of the AF2-predicted ABABAB models revealed unsatisfactory. With the two algorithms, the 14 heteromeric combinations involving a BMC-H with an extended N-ter region were either not predicted to form PF00936 hexamers or were organized as AAABBB hexamers. The only exception was the combination of the two (EutS/PduU), which led to an atypical ABBAAB model proposition by ESMFold, or to an ABABAB with AF2 displaying good quality metrics. These results are not consistent with the moderate to good fluorescence measurements for 5 out of the 14 combinations.

Despite noticed discrepancies depending on the assay, the overall conclusion is that cross-interactions between homologs from different BMC are structurally feasible.

### Cross-linking data support presence of hetero-hexamers in BMC shells

Abovementioned results disclosed a strong degree of structural compatibility between *Kpe* BMC-H homologs, extending conclusions established for carboxysomal components to catabolic BMC [29,30]. All these studies have been carried out

recombinantly. It becomes now mandatory to ascertain whether such mixed associations are integrated in BMC shells in natural contexts. In that respect, a valuable source of information was found within mass spectrometry data reported by Trettel *et al.* [41], which we reinterpreted in the light of our results. In this study, *S. enterica* PDU microcompartments were purified from *E. coli*, after transfer of the whole operon. Lysine residues lying sufficiently close in the shell context were then identified using bifunctional N-hydroxysuccinimide (NHS) crosslinkers coupled to mass spectrometry analysis. Among the many contacts detected in this impressive study, those implicating PduA Lys90 (PduA[90]) and either PduJ[36] or PduJ[89] called our attention. As the authors recognized, such reactions were unanticipated, since K90-K36 and K90-K89 closest couples would be too distant if PduA and PduJ assembled together as adjacent homo-hexamers (*ca.* 43 and 34 Å between primary amine nitrogen atoms, respectively, Fig 7A), since the two reactive NHS moieties of the cross-linker were connected by a 12.5 Å-long arm. To explain these reactions, authors speculated on the possibility that the short PduA C-terminus was flexible, supposedly revolving towards the (PduJ) neighboring hexamer. The high mobility in molecular

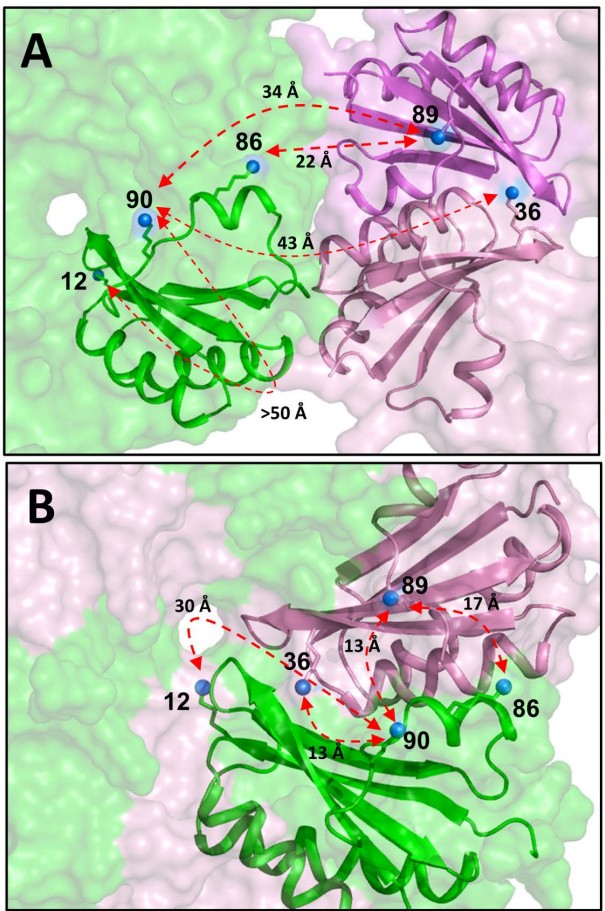

**Fig 7. Cross-linking mass spectrometry data support PduA/PduJ hetero-hexamerization. A.** Localization of cross-linked residues in a hypothetical assembly formed by homo-hexamers of PduA (green) and PduJ (pink, violet). Only one PduA and two PduJ neighboring monomers are shown in cartoon representation. Lysine side-chains are depicted as sticks, blue balls indicating the position of their reactive amino group. Depicted distances are between side-chain nitrogen atoms. **B.** Similar representation to illustrate a hypothetical PduA/PduJ hetero-hexamer, as predicted by AF2. Measured distances are compatible with the detection by mass spectrometry of PduA[90]/PduJ[36] and PduA[90]/PduJ[89] crosslinked peptides. Regardless of whether homo or hetero-hexamer formed, the reaction between the K12/K90 couple of PduA might imply the transfer of a NHS moiety through the central pore, rather than a long-distance displacement towards the hexamer edge.

dynamics (MD) simulations of the C-terminal portion of CcmK2 proteins [57], and the detection of a reaction between PduA[12] and PduA[90], which lie at opposite sides of the hexamer, were presented in support of the argumentation.

From our point of view, it seems unlikely that PduA[90] (and the preceding region) could be as flexible as to cross-react with the K12 primary amine following displacements towards the hexamer edge. The probe would have to transpierce the BMC shell, and reactions with K55 and K72, which lie closer to the edge than K12 on the convex face, should have been detected. Several additional arguments might contribute to discredit the argumentation of Trettel *et al.* (see S11 Fig), including the analysis of local movements during MD simulations on *S. enterica* PduA trihexameric assemblies, which were performed before with a different purpose [58]. We believe, instead, that reported cross-links would be easier to justify within a framework with PduA/PduJ hetero-hexamers incorporated in PDU shells too. As illustrated in Fig 7B, distances of 13–17 Å would be fully compatible with the cross-linker probe radius, not only for PduA[86]-PduJ[89] or PduA[90]-PduJ[89], but most notably for PduA[90]-PduJ[36]. On the other hand, explaining the PduA[12]-PduA[90] cross-reaction would remain similarly challenging, independent of whether homo- or hetero-hexamers occurred. Most reasonably, one of the NHS moieties of the cross-linker might be transferred through the central pore, following K90 displacements in that direction (see S11 Fig).

In the same study, cross-links were also identified between PduJ and PduK or PduA and PduK residues. Such reactions were dominated by K144 or K156 that compose/surround the predicted Fe-S cluster-harboring domain at the end of PduK C-terminal extension. The only mapped PduK-PduK cross-reaction also implicated these two residues. Unfortunately, the preceding portion of this long tail is predicted fully unstructured, thus precluding structural interpretations. In spite of that, the lack of identified reactions on any of the 6 other PduK lysines, some of them corresponding to positions that were reactive in PduA or PduJ, might support an absence of PduK homo-oligomers in BMC shells, indirectly suggesting an integration in the form of hetero-associations.

## Discussion

Understanding BMC assembly and function is fundamental. This particular natural mode of organization contributes to essential biological processes like carbon fixation. Besides, BMC provide new catabolic opportunities to about 20% of bacterial species [11]. Genomic surveys evidenced the existence of a great variety of BMC-mediated processes, something of special interest for synthetic biology purposes. A hallmark of catabolic BMC is the encapsulation of pathways that imply volatile or toxic (aldehyde) intermediates. Accordingly, the major role attributed to the shell is that of a chemical insulator [59]. In this respect, a few experimental studies indicated that shell permeability is defined by the distribution of amino acid residues that surround the central pores of shell constituents [31,32,60], something also supported by theoretical simulations [33,57]. A good example is the S40H mutation of *S. enterica* PduA, which increased the apparent sensitivity of the encapsulated diol dehydratase to inhibition by glycerol, when compared to wild-type BMC [32], which evidenced a more permissive access of glycerol to the BMC lumen. This mutation was selected for being present in a PduA subgroup from PDU compartments that are capable of processing glycerol as an alternative substrate. These studies prove the importance of characterizing hetero-association occurrence in BMC shells, as their physicochemical properties will differ from those of homomers.

Several lines of evidence suggested that hybrid subunits might assemble in BMC shells. On one hand, the multiplicity of shell subunit paralogs inventoried for BMC operons. Thus, about 78% of the 68 BMC types classified by Sutter *et al.* have several BMC-H [11]. On average, 2.6 BMC-H copies were found per BMC *locus*, 2.2 after excluding those with permuted secondary structure elements. Furthermore, arrangements of gene paralog neighbors are not rare. On the other hand, data presented here for catabolic BMC, or before for carboxysomal CcmK from different origins [29,30], proved the formation of hybrid subunits when different BMC-H were simultaneously expressed. Although these studies were performed in *E. coli*, and varied mechanisms could mitigate the extent of hetero-association occurrence in the natural context (see below), demonstrated structural compatibilities hold true. Results from tGFP fluorescence assays presented

here consistently revealed a strong structural promiscuity between BMC-H. Although not all interactions were confirmed in co-purification experiments, a quality indicator was that strongest $F_{max}$ values and lowest sensitivity to tagging schemes occurred for combinations between canonical BMC-H. Strong signals also manifested when canonical and C-terminal extended BMC-H were co-expressed together, in agreement with expectations drawn from sequence identities. Hetero-hexamers consisting of combinations between CmcA/CmcB/CmcC/CmcE or between PduA/PduJ/PduK were clearly pinpointed by the tGFP screen, in relatively good accordance with the structural models predicted by ESMFold or AF2. Some of these combinations were also confirmed to co-purify. Corresponding associations behaved in solution much like homo-hexamers, although basically remaining in soluble fractions after cellular lysis, pointing to weaker macro-assembly trends. Experimental results also supported structural compatibilities between homologs from different BMC types, such as EutM/PduJ or combinations between CmcA, CmcB, CmcC or CmcE with PduA or PduJ, conclusions that were also inferred by the two AI-predictive approaches. Indeed, a remarkable resemblance emerged when considering such predictions (S6 and S7 Tables). Thus, almost invariably, combinations considered directly by ESMFold as incompatible interactors, or leading to AAABBB clusters, also led to AAABBB associations with AF2. In that manner, the two algorithms ruled out any entity implying EutS or PduU. These combinations frequently resulted in low fluorescence readings or signals below negative thresholds for a majority of the 8 screened bicistronic configurations tested. Moreover, none of the 6 selected combinations that included PduU and/or EutS convincingly demonstrated a co-purification of the two partners. In several occasions, however, structural propositions seemed erroneous, like for instance for *Syn6803* CcmK3 or *Kpe* EutK, considered to form homo-hexamers by AF2 or by both AF2 and ESMFold, respectively, or for *Kpe* EutS, which according to ESMFold would not associate into homo-hexamers.

In spite of demonstrated structural compatibilities, several phenomena might prevent hetero-association occurrence in physiological contexts. The first one operates at the transcriptional level, and is presumed to be especially important in preventing the occurrence of chimeric and/or aberrant BMC constituted by mixtures of components from different BMC types. Indeed, several studies not only proved that assembly of chimeric BMC is feasible [61,62], but also that properties like shell permeability, and consequently metabolic function, would be altered [42,60]. Regions inside operons might also be subject to regulation, for instance by means of 2-component regulatory sensors and/or riboswitches like those controlling *eut* and *pdu* expression in *E. faecalis* [63] or *L. monocytogenes* [44]. According to RT-qPCR data, only *grm2* genes were activated among all three BMC in the presence of their cognate substrate in *Kpe*. To a lower extent, CL also induced the transcription of *pdu* and *eutM* regions. To our surprise, EA and PD were without effect on their respective metabolomes. Transcription changes only manifested when combined with CL. On one hand, EA repressed considerably the CL-activation of *grm2,* resembling the repression by PD on EA-activated *eut* transcription (and concomitant *pdu* activation) in *S. enterica* that was demonstrated to prevent deleterious effects caused by the appearance of hybrid BMC [42]. On the other hand, PD combined with CL activated *Kpe pdu* transcription, while still maintaining relatively high *grm2* levels. Accordingly, GRM2/PDU hybrid structures might occur. This could signify that co-expression of some compartments is less detrimental than presumed, at least in some organisms.

At a second level, structural pairing efficiency might be decoupled as a consequence of the chromosomal separation between paralogs. If this parameter was determinant, only demonstrated cross-associations between CmcA/CmcB/CmcC or between PduJ/PduK might be physiologically meaningful in *Kpe*, as other BMC-H genes are distant from any other paralog. Yet, other factors might be the presence of structured mRNA regions or the density of ribosomes simultaneously acting on each coding segment [64]. At the protein level, formation of hetero-hexamers might not necessarily guarantee an integration within BMC shells. In fact, our data pointed to a decline of assembly trends for BMC-H hetero-associations, when compared to homo-hexamers. This could reflect the transmission of small structural deviations towards the hexamer edges, which would be caused by a slightly altered packing of residues at the interface between monomers.

In spite of all mentioned mechanisms, compelling arguments support that hybrid compartments may form. We showed that transcription of *Kpe* BMC operons might be less tightly regulated than in *S. enterica*. Similarly, bile salts were reported

to augment transcription of both *pdu* and *eut* operons in adherent-invasive *E coli* strains [16]. The combination of EA with Vit-B$_{12}$ also increased *eut* and *pdu* readouts in commensal *L. brevis* ATCC 14869 [18]. BMC structures formed *in vivo*, although RT-qPCR readings suggested that shells might be basically PDU-like. The characterization of hybrid PDU/EUT compartments when natural transcriptional regulation was bypassed in *Salmonella* would also suggest that the second and third levels of control might be inoperative [42]. Yet, hybrid compartments might integrate only homomeric subunits from each BMC type. In fact, the most straightforward argument supporting an incorporation of hetero-associations within BMC shells was provided by MS cross-linking data collected for purified *S. enterica* PDU compartments that were produced in an *E. coli* strain harboring the full *pdu1a* operon [41]. We demonstrated that signals implicating cross-reactions between several couples of PduA and PduJ lysines could be easily justified in the context of a *S. enterica* PduA/PduJ mixed oligomer. In line with this, we observed that *Kpe* PduA/PduJ was hetero-hexameric in solution and continued to assemble inside cells. These data are complementary to the likely observation by cryo-EM of CsoS1A/CsoS1B hetero-hexamers in a study of synthetic recombinant α-carboxysomes from *H. neapolitanus* that was published at the time of submission of this work for publication [65].

In conclusion, our study demonstrates a strong structural compatibility between BMC-H and presents compelling evidence in favor of the occurrence of hetero-associations in BMC shells. Future studies will be necessary to evaluate the extent and physiological importance of this phenomenon in natural contexts.

## Materials and methods

### Quantification of RNA transcripts by RT-qPCR

Precultures of *Klebsiella pneumoniae (strain 342)* were prepared from single colonies grown anaerobically in lysogeny broth (LB), overnight (ON) at 37 °C. Final cultures were grown statically in glass tubes (15 × 150 mm) under anaerobic conditions at 37 °C for 48 hours by inoculation of precultures (40 µL) in modified no-carbon-E (NCE) medium (10 mL) [66], priorly degassed for 3 days under nitrogen atmosphere in an anaerobic chamber (PureEvo T4, Jacomex). The modified NCE medium was composed of: 0.82 mM MgSO$_4$, 57.4 mM K$_2$HPO$_4$, 16.74 mM Na(NH$_4$)HPO$_4$, 175 mM NaCl, 20 mM NH$_4$Cl, 50 µM ferric citrate and 5 mg/L thiamine. It also included 0.2% (w/v) yeast extract, 0.0015% (w/v) each of valine, isoleucine, threonine, and leucine, 100 nM cyanocobalamin (vit. B$_{12}$, ref V2876), 50 mM disodium fumarate and trace metals (0.3 mM CaCl$_2$, 0.1 mM ZnSO$_4$, 0.045 mM FeSO$_4$, 0.2 mM Na$_2$Se$_2$O$_3$, 0.2 mM Na$_2$MoO$_4$, 2 mM MnSO$_4$, 0.1 mM CuSO$_4$, 3 mM CoCl$_2$, and 0.1 mM NiSO$_4$). When indicated, the next metabolites were added separately or combined (final conc.): EA hydrochloride (30 mM, pH 7.0, Sigma-Aldrich 411000); PD (55 mM, Sigma-Aldrich 82280); or CL chloride (1%, Sigma-Aldrich, C7017). An equivalent of 3 units of OD$_{620nm}$ were pelleted by centrifugation at 6000 g for 5 min at 4 °C. Pellets were resuspended in 0.5 mL of PBS and 1 mL of RNA*later*® solution (Thermo Fisher Scientific) to avoid transcriptional changes and RNA degradation. Samples were stored at -80 °C until RNA extraction. Total RNA was extracted as described [67]. The absence of DNA contamination was verified by qPCRs performed with the primer pair qPCR-proC-Fw/qPCR-proC-Rv (S8 Table) and the SsoAdvanced SYBR® Green Supermix (Bio-Rad, Hercules, California, USA), according to the manufacturer's recommendations. Reverse transcription was performed with 500 ng of total RNA using the iScript cDNA Synthesis kit (Bio-Rad) following provided indications. qPCRs were carried out in the CFX96 Real Time System (Bio-Rad) with the SsoAdvanced SYBR® Green Supermix (Bio-Rad) in 10 µL total volume per well with 2 µL of 10X diluted cDNA. Primers targeting different *Kpe* genomic regions are listed in S8 Table. Melting curve analysis was used to verify the specific single-product amplification. The amplification efficiency (E) of each primer pair used for the quantification was calculated from a standard amplification curve obtained by five dilution series of genomic DNA. All assays were performed in technical triplicates with three independently isolated RNA samples. Relative quantifications were determined with CFX Maestro software (Bio-Rad), following provider's instructions. The gene expression levels were normalized relative to the expression level of the *proC and rpoD* housekeeping genes [68]. Thus, the relative quantities of all target gene expression were compared to those of the control condition (a value of 1); the expression values are given as fold changes relative to

the condition without substrate (S1 Table). As each condition comprised 3 independent biological replicates (corresponding to a biological group), the expression values shown in Table 1 were calculated by displaying the average expression values of the samples within the biological group. P-values were the results of unpaired *t*-tests comparing the distributions of per well normalized expression values (fold changes) for the control sample *versus* the test sample.

**Preparation of plasmid constructs for recombinant studies**

Nucleotide sequences of the 11 BMC-H of *Kpe 342* (GenBank CP000964.1) were codon-optimized for *E. coli* while maintaining codon frequency profiles along the sequence. For studies of expression of individual BMC-H, resulting DNA sequences, connected to segments coding for $His_6$ tags, were synthesized and provided cloned between NdeI/XhoI sites of a pET29b ($Kan^R$) vector (Twist). Plasmids permitting expression of $His_6$/FLAG BMC-H combinations as bicistrons were also prepared by Twist, cloned similarly in the same vector. However, it was necessary to adapt some codons to permit DNA synthesis. DNA and proteins sequences are compiled in two sheets in an excel file (S9 Table).

The preparation of the library of BMC-H pairs for tGFP screens was carried out following a described protocol [39]. Each codon-optimized BMC-H sequence was synthetized by Twist as DNA fragments in 4 different conformations: GFP10-linker29-BMC-H (N-10 vector), BMC-H-linker31-GFP10 (C-10 vector), GFP11-linker29-BMC-H (N-11 vector) or BMC-H-linker33-GFP11(C-11 vector), with flanking sequences necessary for individual assembly in a pET26b ($Kan^R$) vector opened with XbaI/XhoI. After being PCR amplified (see S10 Table for DNA sequences of primers, BMC-H and vector DNA), each couple of fragments was Gibson-assembled into a BglII/HindIII-opened tGFP vector (pET26b, $Kan^R$) using robotic means provided by a local platform (Toulouse White Biotechnology) (schematized in S4A Fig). Transformation was performed directly in T7 express competent cells (NEB), following the provider protocol. After robotic plating on LB agar (40 µg.mL$^{-1}$ Kan), cells were stored for 2–3 days at 4 °C. Fluorescent clones were then selected on a QPix 460 (Molecular Devices), and the sequence of corresponding plasmid preparation confirmed to be correct.

**Biochemical characterization of individual BMC-H or combinations**

After transformation of BL21(DE3) cells with corresponding pET29 vectors, expression of $His_6$ tagged BMC-H or combinations of $His_6$- and FLAG-tagged BMC-H was carried out in ZYM-5052 auto-induction medium (8 mL) ON at 37 °C, following a described protocol [29]. Cells pellets collected by centrifugation (4000 g) were lysed with Lysis solution (1 mL): BugBuster extraction reagent (Sigma) supplemented with lysozyme (0.03 mg/mL final conc.), Benzonase Nuclease (Merck, 268 U/mL), and PMSF (1 mM). For some experiments, βME (0.75 mM) was also included. Small aliquots were withdrawn and prepared for SDS-PAGE analysis (denatured at 95 °C), corresponding to cellular fractions. Soluble fractions were prepared similarly from material remaining in supernatants after centrifugation at 21000 g, 4 °C for 15 min (pellets were kept at 4 °C for studies of protein disassembly). Purification was performed on Vivapure 8 96-well cobalt-chelate microplate columns (VivaScience), following provider instructions. After 4 washes with 500 mL of solution A (20 mM NaPi/300 mM NaCl/10 mM imidazole, pH 8.0), bound proteins were eluted with 300 µL of a 300 mM imidazole in solution A. EDTA (5 mM) and βME (5 mM) were added immediately after elution and purified fraction aliquots prepared for SDS-PAGE analysis. Material remaining in pellets were resuspended in Lysis solution (1 mL), sonicated for 20 sec at 4 °C and centrifuged at 21000 g. After discarding supernatants, 1 M urea in solution A (1 mL, ±2.5 mM βME) was added to each pellet. After vigorous resuspension, the solution was shaken for 30 min at 4 °C. After 21000 g, urea-solubilized fraction aliquots were prepared for SDS-PAGE as other fractions, and the material was purified as indicated above on microplate columns, leading to urea-purified fractions. Samples were run on 17–18% polyacrylamide gels, which were stained with Coomassie Brilliant Blue R-250 (Bio-Rad).

Cellular and soluble fractions corresponding to the analysis of individual GFP10/GFP11 constructs were prepared as indicated above for $His_6$-tagged BMC-H. Studies were performed with BL21(DE3) cells transformed with N-10, C-10, N-11 or C-11 vectors. The only difference was that ON cultures were grown in LB (2 mL) supplemented with IPTG (10 µM) from the start. βME was omitted during treatments.

Western blots of different fractions were performed as described in [29], after running samples on 17–18% polyacrylamide gels. For experiment presented in Fig 6, an identical second western blot was performed on purified fractions replacing the selected secondary antibody by an antimouse-HRP (Thermo Fisher Scientific, A16066) and the development reactant by a chemiluminescent substrate (TS, ref 34577).

### Tripartite GFP assay

An ON preculture (2 µL) of each tGFP vector-transformed T7 express strain was seeded in Luria-Bertani broth (LB, 200 µL, 40 µg.mL$^{-1}$ Kan), dispensed in a 96-well glass-bottomed black plate (Greiner). IPTG (10 µM, final conc.) was added from the start. Fluorescence signals and optical density at 600nm were monitored continuously on a CLARIOstar Plus (BMC Labtech), as described in [39]. Data were processed with GraphPad Prism 6: the signals of fluorescence obtained were fitted to a sigmoidal function of equation:

with $t_{half\ Fmax}$ the time necessary to reach half of the $F_{max}$ value. The $F_{max}$ values for cases displaying inadequate fits (generally low signal data) were extracted manually. Values were normalized by the $F_{max}$ measured for C-terminally-tagged RMM/RMM in the same assay. Plotted values are the mean $F_{max}$ and ± standard deviations deriving from at least 2 independent repetitions of experiments performed on 3 clones per library member.

### Size-exclusion HPLC

Protein sizes were estimated by SEC using a Beckman Ultraspherogel SEC2000 column (7.5 x 300 mm) mounted on a Waters 2690 HPLC separation module. Purified proteins (100 µL) were dialyzed for 5h (4 °C) in Pur-A-Lyzer dialysis columns (3500 Da cut-off, ThermoFisher), against 500 mL of a 10 mM Tris/200mM NaCl/0.5mM EDTA/pH 7.8 solution. Samples (10–20 µL) were injected at 1 mL/min flowrate after conditioning the column in 20 mM Tris-HCl, 300 mM NaCl at pH 7. Elution was monitored with a Waters 996 Photodiode Array Detector. Elution volumes (280 nm absorption peaks) were used to estimate protein MW by comparison to next calibration standards run under identical conditions: Ferritin (440 kDa), Aldolase (158 kDa), Conalbumin (75 kDa), Ovalbumin (43 kDa) and Ribonuclease (13.7 kDa).

### Transmission electron microscopy

Cells permitting the over-expression of His$_6$-tagged BMC-H, in combination or not with a FLAG-fused partner, were precultured ON at 37 °C in LB (Kan). Next morning, 40 µL of precultures was seeded in 4 mL of LB (Kan). Induction with 200 µM IPTG was triggered in exponential phase [OD (600 nm) of 0.6]. Cultures were continued for 6 h longer at 37 °C (200 rpm). After 1000 g centrifugation, and discarding supernatants, cellular pellets were gently resuspended in 1 mL of fixation mixture: 2,5% glutaraldehyde and 2% paraformaldehyde in cacodylate buffer (0.1 M, pH 7.2). After 15 min, the cells were sedimented again (800 g) and the pellet resuspended in 2 mL of the fixation mixture, which was kept at 4 °C ON. Next morning, the cells were washed in 3 cycles of pelleting (800 g) and gentle resuspension in 1 mL of cacodylate buffer.

Fixed cells were post-fixed with 1% OsO$_4$ in cacodylate buffer. Three washings were performed before inlaying the cells in 2% low-melting point agarose. Then, samples were treated with 1% uranyl acetate for 1h. They were dehydrated using an ethanol gradient: 25, 50, 70 and 90% for 15 min, plus 3 times 30 min at 100%. They were then transferred in Epon resin baths (Embed 812, EMS) of increasing concentration (25, 50, 75% Epon in ethanol for 1 h and twice 2 h in 100% Epon at 37 °C). Finally, they were embedded in Epon resin by a 48 h polymerization at 60 °C. An ultramicrotome UCT (Leica) served to prepare 80 nm-thick sections of the embedded cells. The sections were then mounted onto formvar/carbon-coated copper grids of 200-mesh and stained with Uranyless (EM-grade.com) and Reynolds lead citrate 3% (EM-grade.com). TEM acquisitions were made on a JEM-1400 electron microscope (JEOL Inc, Peabody, MA, USA) operating at 80 kV, equipped with Gatan Orius or Rio 9 digital cameras (Gatan Inc, Pleasanton, CA, USA).

**AF2 and ESMFold predictions**

3D structures of homohexamers and heterohexamers formed by combinations of BMC-H proteins were predicted using two AI-based protein structure prediction tools: the ColabFold implementation of AlphaFold2-Multimer (AF2) version 3 and ESMFold version 1 [51,52]. For AF2-based predictions, the default mode based on MSA, in the absence of template information, was used. For each combination, 20 structural models were generated and the top-ranked model was selected for further analysis. ESMFold, which does not rely on MSA but predicts proteins directly from primary sequences, was used to generate a structural model for each homomeric and heteromeric assembly. For all structural models, the fold, hexameric geometry, and, for heteromeric assemblies, the organization between the two types of monomers were examined using Pymol. In addition to common protein prediction quality metrics provided by AlphaFold and ESMFold such as pLDDT (reflecting local structural accuracy), specific scores dedicated to multimeric assemblies were considered: the average PAE of all interchain residue pairs (interchain_PAE or ic_PAE), the average PAE or pLDDT of interface residue pairs within a 4 Å distance cutoff (interface_PAE, interface_pLDDT). Furthermore, two additional AF2 metrics were provided: the global or interface predicted-TM scores (pTM or ipTM), which asses the reliability of the predicted structure of the complex at a global level or specifically in the interaction interface between complex components. A high-confidence model is characterized by low interface_PAE and interchain_PAE values and high interface_pLDDT values, pTM and ipTM values. In addition, for BMC-H with C- or N-terminal extensions, the interchain_PAE was also computed by excluding these regions (core_PAE).

Relaxation of the structural models was performed using the AMBER protocol from ColabFold followed by the Rosetta FastRelax protocol [69], based on the beta_nov16 energy function, to resolve unfavorable local geometries or clashes. Following relaxation, binding energy scores (ΔE) were computed using Rosetta's InterfaceAnalyzer.

## Supporting information

Supplementary S1-S11 figs and S1-S8 Tables are provided, as well as two excel files with all DNA and protein sequences (S9 and S10 Tables). A supplementary S1 File is also given which contains all averaged $F_{max}$ values presented in Fig 4, as well as accompanying standard deviations and culture times to reach half of $F_{max}$ fluorescence.

**S1 Fig. Comparison of *Kpe* BMC-H sequences. A.** Sequence alignments were prepared with the RCSB pairwise alignment tool (TM-align method), taking as input the structures from the BMC-H monomers generated by AF2. Secondary structure elements from PduA are indicated as reference. EutS and PduU present secondary structure permutations. They were aligned to PduA using the JCE-CP (flexible) method and manually verified. The permutation is highlighted by the black rectangle, indicating the number of the first shown residue. PduU and EutS residues that build the N-terminal β-barrel were excluded. Residues fully or partly embedded at the interface between monomers are indicated by black or grey arrows, respectively. Critical lysine (K26 in PduA) and pore residues that also participate to the interface are indicated by blue and green arrows, respectively. The presentation was generated online with ESPript 3.0. **B.** Percentage of sequence identity between *Kpe* BMC-H. Values based on either residues belonging to the common BMC-H core domain (left) or only those falling at the interface between monomers (right). Coloured values are to highlight cases exhibiting more than 10% discrepancy of identity when comparing the two sets of residues. (TIF)

**S2 Fig. Structural details on predicted BMC-H homo-hexamer models.** Cartoon illustration of portions of representative BMC-H structures predicted by AF2 (left panels) or ESMFold (right panels). From top to bottom are presented the next BMC-H: CmcA (panels *A,B*), EutS (*C,D*), PduK (*E,F*), CmcE (*G,H*) and EutK (*I,J*). The first monomer of the hexamer is colored green, all other monomers are in blue. Views are from the hexamer convex side for CmcA and PduK, concave side for CmcE and EutK. In the case of EutS, a side view is shown. Please note that ESMFold did not predict hexamer for EutS (*D*) and PduK (*F*). The cysteine-rich domain of PduK is highlighted by the dashed red circles, with S-atoms

appearing as orange sticks. To simplify the views, C-terminal extensions in the last three rows are shown only for the first hexamer (cyan). Light green and yellow spheres indicate the localization of the N- or C-terminal residue of the first monomer, respectively. In the middle are included inter-residue pAE score matrices for AF2 structural predictions on the left. Coloring scheme is the one applied by the online Colab AF2 tool (e.g., deep blue for highest confidence level). A to E labels identify each monomer. For each monomer in diagonal elements of PduK or EutK matrices, a first blue box corresponding to BMC-H core residues is followed by small or moderate size blue boxes, respectively, thus supporting high confident folds.
(TIF)

**S3A Fig. Characterization of Kpe BMC-H by size exclusion chromatography.** Profiles after injection of urea purified fractions from indicated BMC-H. EutM and CmcB chromatograms correspond to purified samples (before urea treatments).
(TIF)

**S3B Fig. Characterization of Kpe BMC-H by size exclusion chromatography.** Chromatograms obtained after injection of purified fractions from strains co-expressing indicated BMC-H combinations. Black labels identify the His$_6$-tagged component, in green for the FLAG-partner. N- or C-terminal tag localization is indicated by an asterisk preceding or following BMC-H identity, respectively.
(TIF)

**S4 Fig. The tripartite GFP screening approach. A**. General strategy of construction of plasmids for tGFP assays. Preliminarily, plasmids coding for individual BMC-H with GFP10 or GFP11 tags in either C- or N-terminal were mounted on a pET26b-based template, giving rise to N-10, C-10, N-11 or C-11 vectors. Sequences coding for the two BMC-H of interest were then amplified by PCR from the corresponding plasmids. For simplicity, only one (N-10/C-11) of the eight possible combinations for a given pair of BMC-H is shown here. Primers included 15 nucleotide regions allowing hybridization to either the adjacent fragment (pink box) or to the receptor vector (blue and magenta). The tGFP receptor vector (Kan$^R$) included necessary information for the independent expression of the GFP1–9. The fragments and opened vector were Gibson-assembled giving rise to the final tGFP construct. T7 promoters and terminators are indicated by the arrows and crosses, respectively. **B**. tGFP assay principle: in the case of two interacting BMC-H, the GFP10 and 11 tags will come closer to each other. Reconstitution of a full fluorescent GFP will therefore be promoted in the presence of the GFP1–9 portion. Conversely, GFP reconstitution will be inefficient with non-interacting BMC-Hs.
(TIF)

**S5 Fig. Impact of GFP tag orientation scheme on tGFP signals.** Fluorescence signals deriving from co-expression of Kpe BMC-H homo-pairs with GFP10/GFP11 tags attached following different configurations: N/C, C/N, or N/N orientations. The preparation of plasmid corresponding to the N/N PduK combination failed and could not be assayed (# symbol). Plotted mean F$_{max}$ and standard deviations derive from at least 2 independent repetitions of experiments, each one performed on 3 clones per library member. These values are provided in supplementary S1 File. Other experimental and data analysis details were the same as for Fig 5.
(TIF)

**S6 Fig. Impact of GFP10/11 tag attribution and orientation on BMC-H expression and solubility.** Individual BMC-H either tagged with GFP10 on C- (**A**) or N-terminus (**B**), or with GFP11 on the C- (**C**) or N-terminus (**D**) were overexpressed in BL21(DE3). After recovery of total cellular fractions, centrifugation permitted to prepare fractions corresponding to soluble contents. All fractions were analysed by SDS-PAGE and stained with Instant blue (Expedeon). The approximate migration of ladder components with indicated MW (kDa) are indicated on the left. Black arrows are given as an attempt to identify corresponding BMC-H bands. Theoretical MWs of BMC-H monomer constructs (kDa) are: CmcA

9.4; CmcB 9.6; CmcC 9.5; CmcE 13.6; EutK 17.0; EutM 9.8; EutS 11.6; PduA 9.8; PduJ 9.2; PduK 16.2; PduU 12.4. To these, it is necessary to add the contribution of linkers and tags, which differ as follows: for N-ter GFP10, 4.4; for C-ter GFP10, 4.6; for N-ter or C-ter GFP11, 4.7.
(TIF)

**S7 Fig. Structural details predicted for GRM2 hetero-hexamers.** *A*, A potential disulfide bridge between residues Cys24 and Cys80 of CmcA (geen) and CmcE (violet), respectively, might stabilize contacts between monomers in hetero-hexameric associations. Depicted model corresponds to the AF2 prediction, but a similar result was obtained with ESMFold. Only two neighboring monomers are shown in cartoon representation to facilitate visualization. Cysteine side-chains are depicted as sticks with sulfur atoms as orange balls. Similar arrangement of cysteines is found when CmcA is replaced by CmcB or CmcC. *B*, magnified views of the region around CmcA[24]/CmcE[80].
(TIF)

**S8 Fig. Verification of *Kpe* BMC-H heteromerization.** All details are as in Fig 6, with the exception that fractions were treated in the absence of β-mercaptoethanol.
(TIF)

**S9 Fig. BMC-H heteromers assembly.** TEM images showing contents of *E. coli* cells after over-expression of hetero-pairs combining PduJ-FLAG/PduA-His$_6$ (*A*) or CmcA-FLAG/CmcC-His6 (*B*). Regions displaying nanotubes are indicated by the arrows.
(TIF)

**S10 Fig. Heteromeric association of BMC-H proteins arising from different BMC types.** Please refer to Fig 5 for details on data presentation.
(TIF)

**S11 Fig. Evaluation of the structural flexibility of the PduA K90 region. A.** Comparison of the disposition of K12 and K90 side-chains in deposited 3D structures of PduA variants does not support big displacements towards the hexamer edge required to explain cross-links with K36 and K89 from neighboring PduJ hexamers. Instead, C-terminal residues (including K90) are most often modelled towards the central pore. Wild-type PduA (3NGK) is shown in black trace, in green for all mutants (4PPD, 4QIE, 4QIF, 4GIG, 4RBT, 4RBU, 4RBV). Terminal K12 and K90 amine groups appear as spheres. **B.** Comparison of normalized crystallographic B temperature factors for Cα atoms in the different structures. The blue trace is for 3NGK, black for PduA mutants and green for two CcmK2 structures discussed by Trettel *et al.* (4OX7 and 21AB)[41]. Values were averaged over equivalent modeled residues in the hexamer. Normalized B factor values seem indeed comparable between PduA K90 and the corresponding Arg93 of CcmK2 (dots, following the same color codes). The region preceding this position is not highly mobile, according to MD simulations discussed in ref [41]. **C.** Molecular dynamic analysis does not support K90 extensive movements. Plotted are the evolution of distances between K12/K90 side-chain amine atoms from each monomer during 20 ns run simulations of a PduA trihexameric assembly. Values never differed by more than 10 Å from the initial position, which is that from the crystal structure (3NGK). Shifts indicating approaches were due to movements towards the central pore. For each snapshot, 18 measurements are presented, corresponding to all the monomers from the 3NGK PduA tri-hexameric assembly studied before [58]. Note that inter-amine distances are measured in straight line, which is still incompatible with reaction in most instances.
(TIF)

**S1 Table. Induction of *Kpe* BMC transcription in response to metabolite presence.** [a] Expressions values are calculated from three independent replicates, and are given as fold changes relative to the condition without substrate.[b] SEM

values correspond to the expression lower and upper error bars. Control measurements of housekeeping gene expression are compiled in S2 Table.
(PDF)

**S2 Table. Reproducibility of transcription level measurements of housekeeping genes.** [a] mean quantification cycle, determined from three technical replicates.
(PDF)

**S3 Table. Evaluation of the quality of homo-hexamer models predicted by AlphaFold2 and ESMFold.** [a] Successful prediction of PF00936 hexameric associations is specified as YES/NO. [b] Global pLDDT. [c] interchain PAE, between pairs of residues belonging to different chains. [d] Values between pairs of interchain residues lying closer than 4 Å from each other. [e] values for interchain residues belonging to the BMC-H core, only calculated for non-canonical BMC-H. [f] AF-multimer pTM and ipTM scores below 0.7 indicate low confidence prediction quality. [g] Interaction energy ($\Delta E$) computed using Rosetta InterfaceAnalyzer after structure relaxation, and averaged over all interface$\Delta$s. Values of if_pLDDT, if_PAE and $\Delta E$ are reported only for combinations predicted as PF00936 hexamers.
(PDF)

**S4 Table. Evaluation of inter-monomer interaction energies from experimental 3D structures.** [a] Averaged over all interfaces present in the 3D structure.
(PDF)

**S5 Table. Assessment of *Kpe* BMC-H oligomeric state by SEC-HPLC.** [a]Theoretical molecular weight (MW) of monomer, in kDa. [b] MW estimated from peak elution volume, in kDa. Labels 1-M and 2-M are to indicate first or second major species in intensity, whereas m is for minor or faintly detected species; ND for nothing detected within the 5–500 kDa resolving range of the column. [c] These samples revealed peaks eluting as high-MW soluble species of highest intensity (1-M). Their retention time was between those of ferritin (440kDa) and dextran blue (2MDa), or even above the latter. Asterisks indicate tag attachment to the BMC-H N-terminus.
(PDF)

**S6 Table. Analysis of ESMFold and AF2 predictions for hetero-hexamers combining monomers from the same BMC type.** [a] Organization mode of monomers in the top ranked predicted model. A and B are to identify the first and second monomer types. [b] Energies are averaged over the two different sets of 3 identical interfaces from the ABABAB hexamers. Other details are as in S3 Table.
(PDF)

**S7A Table. Analysis of AF2 predictions for hetero-hexamers combining monomers from different BMC types.** Data were organized as in S6 Table.
(PDF)

**S7B Table. Analysis of ESMFold predictions for hetero-hexamers combining monomers from different BMC types.** Please refer to S6 Table for data organization details.
(PDF)

**S8 Table. Primers used for RT-qPCR experiments.**
(PDF)

**S9 Table. DNA and protein sequences of constructs permitting the expression of Kpe BMC-H homo-hexamers (His$_6$-tagged) or the co-expression of BMC-H pairs (FLAG and His$_6$-tagged).**
(XLSX)

**S10 Table. DNA sequences of fragments, plasmids and primers used in this study for the preparation of constructs screened with the tGFP technology.**
(XLSX)

**S1 File. Average $F_{max}$ values with standard deviations from tGFP screenings of *Kpe* BMC-H pairs.** Values are expressed as percentages of measurements for the RMM/RMM reference case (100%) and correspond to the average of at least 6 measurements, generated in 2 independent experiments. These values were used to prepare Fig 4. Measurements of halt time to reach maximal fluorescence after fits to a sigmoidal function are also provided.
(XLSX)

**S1_raw_images.** Original uncropped and unadjusted images underlying SDS-PAGE gels and blots presented in this work.
(PDF)

## Acknowledgments

We thank Prof. Thomas A. Bobik for helpful discussions concerning *Klebsiella pneumoniae* culturing conditions.

## Author contributions

**Conceptualization:** Luis F. Garcia-Alles, Damien Balestrino, Delphine Dessaux, Thomas Schiex, Sophie Barbe.

**Data curation:** Luis F. Garcia-Alles, Damien Balestrino, Delphine Dessaux.

**Formal analysis:** Luis F. Garcia-Alles, Lucie Barthe, Damien Balestrino, Bessam Azizi, Delphine Dessaux.

**Funding acquisition:** Luis F. Garcia-Alles.

**Investigation:** Luis F. Garcia-Alles, Lucie Barthe, Delphine Dessaux, Vanessa Soldan, Jeremy Esque, Thomas Schiex, Sophie Barbe.

**Methodology:** Luis F. Garcia-Alles, Lucie Barthe, Damien Balestrino, Bessam Azizi, Delphine Dessaux, Vanessa Soldan, Jeremy Esque, Thomas Schiex, Sophie Barbe.

**Project administration:** Luis F. Garcia-Alles.

**Supervision:** Luis F. Garcia-Alles.

**Validation:** Luis F. Garcia-Alles, Lucie Barthe.

**Visualization:** Luis F. Garcia-Alles, Lucie Barthe.

**Writing – original draft:** Luis F. Garcia-Alles, Lucie Barthe.

**Writing – review & editing:** Luis F. Garcia-Alles, Lucie Barthe, Damien Balestrino, Bessam Azizi, Delphine Dessaux, Sophie Barbe.

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
