## [Decision Letter · Decision Letter 0]

27 Feb 2025

PONE-D-25-00799Promiscuous structural cross-compatibilities between major shell components of Klebsiella pneumoniae bacterial microcompartmentsPLOS ONE

Dear Dr. Garcia-Alles,

Thank you for submitting your manuscript to PLOS ONE. After careful consideration, we feel that it has merit but does not fully meet PLOS ONE’s publication criteria as it currently stands. Therefore, we invite you to submit a revised version of the manuscript that addresses the points raised during the review process. Specifically, ensure that statistical significance is reported, where feasible, and pay close attention to the language used in order that the conclusions, as stated, are fully supported by the reported data. Refer to the reviewer comments for additional, specific comments. Note that reviewer 1 did not complete a thorough review of the manuscript and, as such, you may disregard those comments.

We look forward to receiving your revised manuscript.

Kind regards,

Jarrod B. French, PhD

Academic Editor

PLOS ONE

Journal Requirements:

2. Thank you for stating the following financial disclosure: The French National Research Agency (ANR) financially supported this work: ANR-19-CE09-0032-01. This work also benefited from a grant managed by the same agency, under the "Investissements d'Avenir" programme: ANR-18-EURE-0021.

This work was granted access to the HPC resources of CALMIP supercomputing center. 

3. Thank you for stating the following in the Acknowledgments Section of your manuscript: The French National Research Agency (ANR) financially supported this work: ANR-19-CE09-0032-01. This work also benefited from a grant managed by the same agency, under the "Investissements d'Avenir" programme: ANR-18-EURE-0021. This work was granted access to the HPC resources of CALMIP supercomputing center. We thank Prof. Thomas A. Bobik for helpful discussions concerning Klebsiella

pneumoniae culturing conditions We note that you have provided funding information that is not currently declared in your Funding Statement. However, funding information should not appear in the Acknowledgments section or other areas of your manuscript. We will only publish funding information present in the Funding Statement section of the online submission form. 

The French National Research Agency (ANR) financially supported this work: ANR-19-CE09-0032-01. This work also benefited from a grant managed by the same agency, under the "Investissements d'Avenir" programme: ANR-18-EURE-0021.

This work was granted access to the HPC resources of CALMIP supercomputing center.

Reviewers' comments:

Reviewer's Responses to Questions

**Comments to the Author**

1. Is the manuscript technically sound, and do the data support the conclusions?

Reviewer #1: No

Reviewer #2: Yes

Reviewer #3: Yes

2. Has the statistical analysis been performed appropriately and rigorously? 

Reviewer #1: I Don't Know

Reviewer #2: N/A

Reviewer #3: I Don't Know

3. Have the authors made all data underlying the findings in their manuscript fully available?

Reviewer #1: Yes

Reviewer #2: Yes

Reviewer #3: Yes

4. Is the manuscript presented in an intelligible fashion and written in standard English?

Reviewer #1: Yes

Reviewer #2: Yes

Reviewer #3: Yes

5. Review Comments to the Author

Reviewer #1: I cannot review this work by the deadline given, as the formatting for what I was given for review requires too much flipping back and forth to see what I'm looking at, and the SI figures are missing captions altogether in the powerpoint given to the reviewers. The formatting chosen by the authors separates captions from figures, both in the main text and the SI. The main text figures themselves are of low quality, as though they had been through a lossy compression algorithm before I ever got to see them. I cannot make a determination on the work to see whether the conclusions (which seem reasonable based on the text!) are actually supported by the data, as the figures without their captions are insufficient for me to judge what is actually going on. The work itself is *probably* fine, and the conclusions again seem perfectly reasonable based on what I know about BMB and protein structure, but I can't verify the claims easily.

Reviewer #2: This manuscript describes bioinformatic structural predictions of hetero-hexamer formation by bacterial microcompartment shell proteins from Klebsiella pneumoniae and goes on to co-expression studies demonstrating their likely formation in vitro by expression of different labelled protein pairs in E.coli. Because K.pneumoniae contains three different bacterial microcompartment operons there are a large number of possible combinations addressed. While all operons show some inter-operon protein associations, the Cut operon and Pdu operon shell proteins associate more than the Eut operon shell proteins do with proteins from either of the other two operons. There is some RT-PCR regulatory evidence that choline plus another substrate is more of a signal than 1,2-PD or ethanolamine. PduA/PduJ association is particularly favoured in the expression study and the structural prediction is of alternating monomers. The authors use this prediction to explain otherwise puzzling observations of a previous crosslinking study of BMC proteins from a heterologously expressed Salmonella Pdu operon by Trettel at al (1).

It is of interest to apply the bioinformatic tools to the problem of multiple BMC structures in the same organism, and the expression study correlations are useful.

I have the following points:

1. I don't think the application of their findings to Trettel et al' s observations is well expressed. I can't reconcile Figure 7 with Trettel Figure 6D: in Trettel 6D K90-K12 in the homo-hexame PduA/PduJ cross-linked structure is <25 angstroms, in Figure 7A it is >50: K86-K89 is > 30 angstroms in Trettel 6D, 22 Angstroms in Figure 7A. I think Figure 7 needs to be revised to make it clearer how the cross-linking data fits a heterohexamer PduA/PduJ than the Trettel PduA/PduJ homohexamer arrangement

2. Page 24 "constitutive expression of Kpe grm2 in the gut, whereas the PDU or EUT would provide novel metabolic opportunities to the organism under inflammatory contexts"

This seems unlikely, as the authors point out themselves, Lactobacillus brevis is just one of the many non pathogens which have Pdu and Eut operons. 1,2-propanediol is produced by several intestinal organisms from fucose, ethanolamine derives from cell membranes in the diet and turned over in the intestine, they are both readily available for commensal organisms in the intestine. There is no Tetrathionate reductase in Klebsiella for the accelerated respiration of these compounds. The speculation to be made could be - exactly where in the intestine are choline and 1,2-PD , or choline and ethanolamine likely to be ? and does that correspond with K.pneumoniae carriage ?

3. Typos etc Page 20 Line 11 "floopy" for floppy, "flexible" would probably be a better word here anyway.

In a couple of places there are what look like direct translations P15 "all GRM2 BMC-H combinations conducted to strong fluorescence", P20 "Trettel’s et al argumentation "

References

1. Trettel, D. S., W. Resager, B. M. Ueberheide, C. C. Jenkins, and W. C. Winkler. 2022. Chemical probing provides insight into the native assembly state of a bacterial microcompartment. Structure 30: 537-550.e5.

Reviewer #3: recommendation: accept following revision

L. Barth et al. report here on potential variations in the assembly of bacterial microcompartments (BMCs). BMCs are essential for the sequestration of biochemicals producing toxic or highly reactive intermediates, and are canonically considered to be assembled from a single set of protein shell components per associated reaction pathway. In this manuscript, genetic analysis of operons in K. pneumoniae identifies multiple homologous BMC-H genes and multiple types of BMC in a single organism, challenging this assumption. The authors use multiple orthogonal techniques, including in vivo, in vitro, and in silico assays, to assess the combinatorial formation of heterologous BMCs. While the conclusions reflect the somewhat messy and difficult to interpret nature of the complicated system, they are appropriately analyzed, reported accurately, and generally properly justified.

notable strengths

The heart of the paper appears to be figures 4 and 5, which are thoroughly analyzed. Possible caveats with respect to expression, tag interference in folding, and differentiating lack of interaction due to poor behavior from bona fide lack of interaction are appropriately addressed.

notable weaknesses

Details on statistical analysis are scant, but this should be easily correctable and seems unlikely to affect the conclusions of the manuscript.

Conclusions based on structural prediction are not fully supported

specific comments (sorted by figure)

Figure 1 - none

Table 1 and associated supplemental -

Major point: Please provide details of the statistical analysis. In particular, is each triplicate in table S1 treated separately or is ANOVA used within each gene's set of conditions? Ideally, it will be the latter.

Minor point: use of comma or period to indicate the decimal point is inconsistent (and in the supplemental tables in general)

Figure 2 and associated supplemental -

Major point: Related to Table S5, it is difficult to assess the reliability of the SEC analysis without seeing the trace itself. Is a clean peak seen, or is the highest part of a broad spread reported? Please provide in supplemental data. Caveats about certain samples appear to be appropriately noted in the text.

Minor point: according to methods, I believe these proteins had urea removed by dialysis to refold them before SEC analysis? It may be worth briefly noting this in the main text or appropriate legend.

Figure 3 -

Minor point: Ordering of panels by letter is odd.

Minor point: It would be helpful to provide an image of at least one of the constructs that failed to assemble as a negative result against which to compare the less evident assemblies such as panels B and F.

Figure 4 -

Figure S3 is helpful in understanding the experiment.

Minor point: Was tripartite GFP chosen instead of split GFP to minimize the size of the tag attached to the individual BMC constructs?

Figure 5 -

Major point - as with Table S1, please provide details of statistical analysis

Figure S5 -

Major point - I am not convinced that this supports the statement that these tags interfere somewhat with assembly formation, largely because so many of bands are barely visible to begin with (and why does EutK not appear to change in size?). Analysis may be best limited to confirmation of expression. Failure to observe expression of a construct by this method should be highlighted, but merely indicates that the protein is not abundant enough to be seen above lysate baseline; any noise in the data due to poor expression would already be addressed in Figure 5.

Figure 6 -

Appropriately analyzed and discussed in text

Minor point: Bringing the arrows in front of the lane dividers instead of behind, and perhaps making them brighter, may make the figure easier to interpret without zooming far in

Figure 7 -

No comments

Structural prediction -

Major point: discussion of c-terminal extensions depends on their reported local confidence, which given the predictions appears low. Either justify the accuracy of the predictions, possibly by coloring Figure S2 by local confidence, or remove discussion of extension structure from the manuscript.

Minor point: if pTM and ipTM values from AlphaFold-Multimer are available, please add to Table S3

6. PLOS authors have the option to publish the peer review history of their article (what does this mean? ). If published, this will include your full peer review and any attached files.

**Do you want your identity to be public for this peer review?** For information about this choice, including consent withdrawal, please see our Privacy Policy .

Reviewer #1: No

Reviewer #2: No

Reviewer #3: **Yes: ** Brian Compton Richardson

---

## [Author Response · Author response to Decision Letter 1]

20 Mar 2025

A letter with reply to all reviewers' comments is provided along with all other files of this revision

---

## [Editor Report · Decision Letter 1]

25 Mar 2025

Promiscuous structural cross-compatibilities between major shell components of Klebsiella pneumoniae bacterial microcompartments

PONE-D-25-00799R1

Dear Dr. Garcia-Alles,

We’re pleased to inform you that your manuscript has been judged scientifically suitable for publication and will be formally accepted for publication once it meets all outstanding technical requirements.

Kind regards,

Jarrod B. French, PhD

Academic Editor

PLOS ONE

---

## [Editor Report · Acceptance letter]

PONE-D-25-00799R1

PLOS ONE

Dear Dr. Garcia-Alles,

I'm pleased to inform you that your manuscript has been deemed suitable for publication in PLOS ONE. Congratulations! Your manuscript is now being handed over to our production team.

Kind regards,

on behalf of

Professor Jarrod B. French

Academic Editor

PLOS ONE